# Longitudinal tracking of acute kidney injury reveals injury propagation along the nephron

Luca Bordoni [1,2,4], Anders M. Kristensen [1,4], Donato Sardella [1], Hanne Kidmose[1], Layla Pohl [1], Søren Rasmus Palmelund Krag [3] & Ina Maria Schiessl [1] ✉

Acute kidney injury (AKI) is an important risk factor for chronic kidney disease (CKD), but the underlying mechanisms of failed tubule repair and AKI-CKD transition are incompletely understood. In this study, we aimed for dynamic tracking of tubule injury and remodeling to understand if focal injury upon AKI may spread over time. Here, we present a model of AKI, in which we rendered only half of the kidney ischemic. Using serial intravital 2-photon microscopy and genetic identification of cycling cells, we tracked dynamic tissue remodeling in post- and non-ischemic kidney regions simultaneously and over 3 weeks. Spatial and temporal analysis of cycling cells relative to initial necrotic cell death demonstrated pronounced injury propagation and expansion into non-necrotic tissue regions, which predicted tubule atrophy with epithelial VCAM1 expression. In summary, our longitudinal analyses of tubule injury, remodeling, and fate provide important insights into AKI pathology.

Acute kidney injury (AKI) is defined as a sudden decline in renal function, which is often caused by acute tubule injury[1]. Depending on the severity of the insult, kidney function in patients may recover. However, it is by now well established that AKI patients maintain a high risk of developing chronic kidney disease (CKD) later[2,3]. The mechanisms of this AKI-CKD transition are incompletely understood[3] but associated with endothelial cell dysfunction[4,5], interstitial immune cell recruitment and inflammation[6–8], renal interstitial fibrosis[9,10], and incomplete tubule epithelial repair[11,12]. The renal proximal tubule (PT) is the main injury site during ischemia-reperfusion injury (IRI) and undergoes significant remodeling[13]. Upon injury, PT cells can dedifferentiate with de novo expression of markers such as Kim-1, Cd44, and vimentin, proliferate and eventually re-differentiate into functional PT cells[14–16]. Nevertheless, in their dedifferentiated state, they also engage in paracrine signaling pathways enhancing interstitial inflammation, myofibroblast recruitment, and subsequent fibrotic remodeling[14,17–19]. Recent single-cell sequencing studies obtained from IRI kidneys further suggested that a subset of these dedifferentiated

PT cells undergo a state of failed repair with increased inflammatory signaling[8,11]. Lastly, AKI-induced necrotic cell death may engage directly in enhanced tissue inflammation and promotion of further injury[20,21].

To prevent AKI-CKD transition, we need a better understanding of how tissue injury and remodeling processes link to successful or unsuccessful recovery. However, there are limited longitudinal data to study this. AKI patient biopsies are not routinely collected and demonstrate considerable heterogeneity with respect to the time biopsies were taken[22]. Furthermore, animal studies of AKI classically involve organ collection for further ex vivo analysis. Thus, highly dynamic AKI-induced remodeling processes are reconstructed from single snapshots acquired from different animals and time points. This limits insights into dynamic cell–cell interactions and their impact on nephron fate. In fact, several rather basic questions on how renal tissue may regenerate from injury are still unanswered or controversially discussed: Which cells proliferate in response to tissue injury? How big is the recovery capacity of injured renal tissue?

[1]Department of Biomedicine, Aarhus University, Aarhus, Denmark. [2]GliaLab and Letten Centre, Division of Anatomy, Department of Molecular Medicine, Institute of Basic Medical Sciences, University of Oslo, Oslo, Norway. [3]Department of Pathology, Aarhus University Hospital, Aarhus, Denmark. [4]These authors contributed equally: Luca Bordoni, Anders M. Kristensen. ✉e-mail: Ina.maria.schiessl@biomed.au.dk

How is uninjured renal tissue influenced by adjacent focal injury? Can injury spread over time?

In this study, we combined serial intravital 2-photon microscopy (2PM) with genetic identification of cycling cells[23] and a model of partial ischemia-reperfusion injury (partial IRI), which was conceptualized to study the effects of necrotic cells on adjacent uninjured tissue. Serial imaging of partial IRI kidneys over 3 weeks provided paired data of tubule injury and remodeling and, for the first time, linked initial necrotic cell death to local sites of tubule proliferation and eventually tubule fate. Thus, our data identified the early proximal tubule as a key site of necrotic injury and source of granular cast formation, which was associated with downstream injury propagation and development of tubule atrophy.

## Results

### A model of partial ischemia-reperfusion injury (partial IRI)

To study the effects of focal necrotic injury on uninjured renal tissue in AKI over time, we have established a model of partial ischemia-reperfusion injury. Through occlusion of only one branch of the left renal artery, approximately half of the organ was rendered ischemic, while the other half remained perfused (Fig. 1a, d). Reperfusion was granted after 21 min. On day 2 after surgery, partial IRI mice displayed significantly increased urinary albumin excretion (albumin/creatinine ratio (ACR), 24357.5 ± 10073.2 vs. 3333.1 ± 786.3 mg/g, mean ± SEM, $p = 0.001$, $n = 4$ and 3, partial IRI vs. sham, respectively), (Supplementary Extended Statistics.a), which thereafter recovered to sham levels (Fig. 1b). Glomerular filtration rate (GFR) was measured in awake sham and partial IRI mice through assessment of FITC-sinistrin clearance and revealed no differences between the groups over 7 weeks (Fig. 1c).

### Tubular necrotic cell death distribution

For longitudinal follow-up of renal injury and (failed) recovery, an abdominal imaging window (AIW) was implanted above the partial IRI kidney, allowing to repeatedly visualize full ischemic (IR), non-ischemic (Not-IR) areas, as well as the border regions in between (Mid) (Fig. 1d). Two hours after reperfusion, in vivo 2PM of partial IRI kidneys, allowed discrimination of IR, Not-IR, and Mid regions based on distinct spatial distribution of necrotic cell death as detected by nuclear propidium iodide (PI)-staining. PI-positive tubule cells further revealed markedly reduced blue autofluorescence ($\lambda_{Ex} = 750$ nm, $\lambda_{Em} = 435$–485 nm), indicative of NADH (nicotinamide adenine dinucleotide) depletion[24] (Fig. 1e). In IR regions, epithelial necrotic cell death averaged 21.78 ± 1.32% PI-positive nuclei per segment's total nuclei (mean ± SEM, $n = 263$ segments from 6 mice), of which 76.04% of all analyzed tubule segments demonstrated epithelial necrotic cell death of varying severity (Fig. 1f, g). In Mid regions, significantly less injury was detectable (6.21 ± 0.54% necrotic cells, mean ± SEM, $n = 334$ tubule segments from 7 mice, $p < 0.001$, Supplementary Extended Statistics.d) and 48.5% of the segments did not demonstrate any epithelial necrotic cell death. In Not-IR regions, 4.88% of the investigated tubules demonstrated scattered epithelial necrotic cell death of minor extent, but the extent of necrosis was not significantly different from zero (0.15 ± 0.07% necrotic cells, mean ± SEM, $n = 123$ tubule segments from 5 mice, $p = 0.28$, Supplementary Extended Statistics.e), (Fig. 1f, g).

We next investigated the distribution of tubular necrotic cell death after ischemic injury along the nephron. Using 2PM, S1 proximal tubule (PT-S1), S2 proximal tubules (PT-S2), as well as distal convoluted tubules, connecting tubules and collecting duct tubule segments are accessible[25,26], of which the latter three were collectively analyzed (DCT/CD) in this study. NADH and FADH (flavin adenine dinucleotide) play important roles in PT ATP production and are simultaneously excitable at 750 nm using 2PM[24,27]. Consistent with previous data[25], distinct patterns of blue ($\lambda_{Em} = 435$–485 nm, NADH) and green PT autofluorescence ($\lambda_{Em} = 500$–550 nm, FAD) (Fig. 2a, b) allowed reliable subclassification of accessible PT segments into PT-S1 and PT-S2. Thus,

PT-S2 segments displayed significantly stronger green and blue fluorescence compared to both PT-S1 and DCT/CD, and the ratio of green/blue emission light allowed significant distinguishment of PT-S1, PT-S2, and DCT/CD, respectively (Fig. 2b). Furthermore, bolus-tracking during i.v. injection of freely filtered FITC-conjugated 4 kDa dextran in 4 mice demonstrated markedly delayed bolus arrival times in tubule segments classified as PT-S2 as compared to those classified as PT-S1 (Fig. 2c). Out of 875 tubule segments analyzed from partial IRI mice, 8.7% were unclassifiable due to severe necrosis and excluded from classification.

As expected, PT segments were more injured than DCT/CD segments. In IR regions, necrotic cell death in PT segments averaged 20.41 ± 1.49% PI-positive nuclei ($n = 195$ segments from 6 mice), of which 72.82% of all proximal tubule segments displayed necrotic cell death of varying severity. In contrast, necrotic cell death in DCT/CDs of IR regions was overall of low extent compared to PT segments (2.89 ± 0.75% PI-positive nuclei, $n = 14$ segments from 4 mice, $p < 0.001$ vs. PTs, Supplementary Extended Statistics.h) with 57.14% of all DCT/CD segments demonstrating scattered necrotic cells death (<10% PI+ nuclei over total nuclei). When distinguishing between PT-S1 and PT-S2 segments, we observed a significantly higher degree of necrotic cell death in PT-S1 segments (26.19 ± 2.45% PI-positive nuclei, $n = 90$ tubule segments from 6 mice) than in PT-S2 (15.45 ± 1.68% of PI-positive nuclei, $n = 105$ segments from 6 mice, $p < 0.001$, Supplementary Extended Statistics.i) (Fig. 2e, f). Similar results were obtained when analyzing IR and Mid regions collectively and when normalizing the count of PI-positive cells over the total tubule cross-sectional area rather than the total nuclear count (Supplementary Fig. 1).

Necrotic cell death, as detected by PI staining, only occurred within the first few hours after reperfusion. Two hours after reperfusion, most PI-positive cells appeared to be still attached to the basal membrane. However, 6 h after reperfusion, we observed massive casting of necrotic cells into the tubular lumen (Fig. 1h).

### Definition and classification of distinct remodeling events in post-ischemic kidneys using 2PM

When following post-ischemic tubules over time using intravital 2PM imaging, we detected distinct morphological changes of the remodeling epithelium alongside recognizable autofluorescence patterns and materials. For a thorough pathological classification of the observable events, we performed serial cryosections of fixed post-ischemic kidneys ($n = 3$), of which two were stained for hematoxylin-eosin (HE) and periodic acid-Schiff (PAS) staining, respectively. A third serial section was imaged by 2PM using the same settings as in vivo. Using 2PM of cryosections, we identified typical remodeling events, as frequently observed in vivo, and we then re-identified these regions of interest in consecutive HE- and PAS-stained sections using correlative microscopy (Supplementary Fig. 2). Correlated microscopy images from frozen sections as well as in vivo 2-photon images from this study were screened and classified by an experienced pathologist. The following phenomena were classified and further investigated: (1) Luminal accumulation of highly autofluorescent cellular debris in tubules was identified as granular casts and acute tubule necrosis (ATN). (2) Dilated tubules with flattened and simplified epithelium were identified as acute tubule injury (ATI). (3) Collapsed and bright autofluorescent tubules of fragmented appearance were identified as atrophic (Supplementary Fig. 2).

### Dynamic epithelial cell cycling post AKI

To investigate epithelial cell cycling post AKI in vivo and over time, we performed serial in vivo imaging of transgenic CycB1-GFP reporter mice, which indicate cells entering phases S, G2, and M of the cell cycle through transient GFP expression[23] (Fig. 3a, and Supplementary Fig. 5). To validate that CycB1-GFP mice truly identified proliferating cells by GFP expression, we injected EdU daily for 4 days while performing

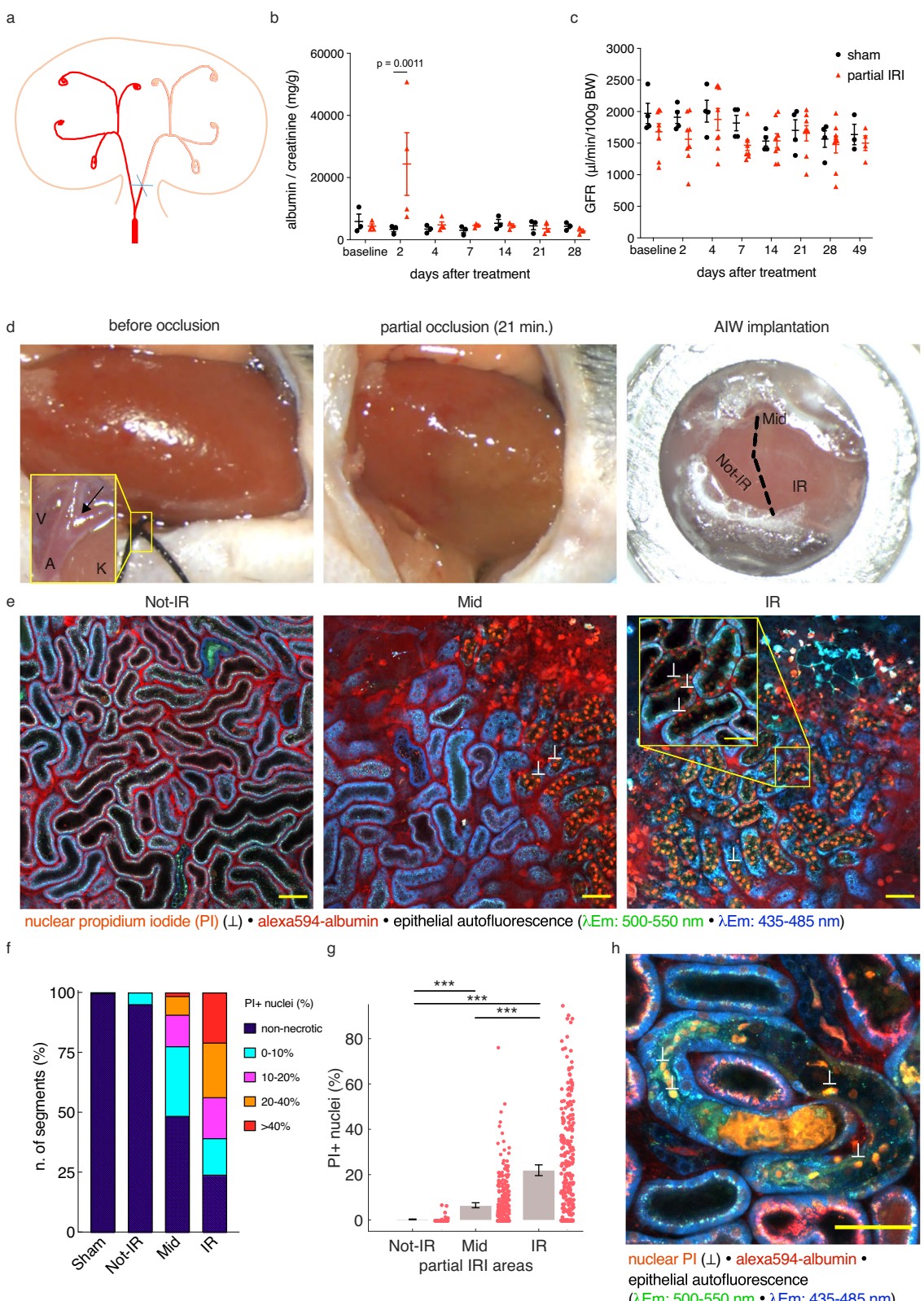

nuclear propidium iodide (PI) (⊥) • alexa594-albumin • epithelial autofluorescence (λEm: 500-550 nm • λEm: 435-485 nm)

nuclear PI (⊥) • alexa594-albumin • epithelial autofluorescence (λEm: 500-550 nm • λEm: 435-485 nm)

in vivo imaging of GFP expression in the kidneys. After perfusion-fixation and ex vivo EdU-labeling, we then performed correlative imaging and confirmed that GFP-expressing cells were also identified as EdU-incorporating cells (Supplementary Fig. 4). We then quantified tubular GFP expression in serially imaged sham and partial IRI CycB1-GFP kidneys. GFP expression in sham kidneys was generally low over a time course of 3 weeks (Supplementary Table 1, Fig. 3b, and

Supplementary Fig. 5a). In partial IRI kidneys, the number of GFP-expressing epithelial cells increased within the first week after reperfusion, peaked at day 3, and was associated with distinct epithelial flattening and loss of tubule brush border, indicative of ATI (Fig. 3a). In week 2 and 3 after reperfusion, epithelial GFP expression in partial IRI kidneys decreased to sham levels (Supplementary Table 1, Fig. 3b, and Supplementary Fig. 5a). We next analyzed GFP expression in IR, Mid,

**Fig. 1 | A model of partial ischemia-reperfusion injury (partial IRI). a** Schematic drawing of the partial IRI model. **b, c** Urinary albumin/creatinine ratio and glomerular filtration rate (GFR) from sham and partial IRI mice (b: $n = 3$ and 4 mice and c: $n = 4$ and 8 mice for sham and partial IRI, respectively); mean ± SEM with scatterplot. Statistical test: repeated measurement two-way ANOVA, factors: treatment and days from treatment, post-hoc analysis: multiple comparisons, Bonferroni correction (Supplementary Extended Statistics.b). **d** Representative stereomicroscope photos of a mouse kidney before and during renal artery branch occlusion for 21 min and after abdominal imaging window (AIW) implantation upon reperfusion. Ischemic and perfused regions were clearly identifiable until reperfusion. AIW implantation was performed on the interphase between ischemic (IR), non-ischemic (Not-IR) regions, and the area in between (Mid). Inlet: Arterial bifurcation (V: renal vein, A: renal artery, K: kidney. Arrow indicates bifurcation). **e** In vivo 2-photon images acquired 2 h after partial IRI, identified IR, Mid, and Not-IR regions. Scale bar: 50 µm. Note marked reduction of blue autofluorescence in propidium iodide (PI)+ tubule cells (inlet, scale bar: 30 µm). **f** Necrotic injury distribution in sham and different injury regions of partial IRI kidneys clustered by percentage of segments with indicated threshold of PI+ nuclei (% of total nuclei/ segment). **g** Volumetric quantification of PI+ nuclei (% of total nuclei/ segment), ($n = 123$, 334, and 263 segments from 5, 7, and 6 mice for Not-IR, Mid, and IR, respectively, and 360 segments from 3 Sham mice); mean ± 95% CI with scatterplot. Statistical test: linear mixed-effect model, $p$-values from two-sided test (Supplementary Extended Statistics.c). **h** In vivo 2-photon image acquired 6 h after partial IRI, displaying luminal accumulation of PI+ cells. Scale bar: 50 µm. *: $p < 0.05$; **: $p < 0.01$; ***: $p < 0.001$.

and Not-IR regions of partial IRI kidneys, respectively, and over time. Two hours after reperfusion, there was no difference between the groups. GFP expression in IR areas started to increase from day 1 and was significantly higher than in Mid and Not-IR regions between days 1 and 4 after reperfusion (Supplementary Table 1, Fig. 3c, and Supplementary Fig. 5b). In contrast, GFP expression in Mid regions increased only from day 2 and remained elevated until day 7 post reperfusion when compared to Not-IR regions. Seven days post reperfusion, GFP expression in IR and Mid regions was comparable but elevated when compared to Not-IR regions. On days 14 and 21 after reperfusion, there was no difference between the groups (Supplementary Table 1, Fig. 3c, and Supplementary Fig. 5b). In Not-IR regions, proliferative activity was generally low. However, on days 2 and 3 post reperfusion, GFP expression slightly increased and reached significantly higher levels compared to sham (Supplementary Table 1, Supplementary Fig. 5a inlet).

Finally, we assessed the overall tubule segment-specific proliferative capacity after partial IRI. One day after reperfusion, we found that PT-S1 segments proliferated more than PT-S2 and DCT/CD segments respectively. However, on days 2 and 3, epithelial GFP expression in PT-S2 segments drastically increased and exceeded that of PT-S1 and DCT/CD, respectively. There was no difference between the groups on days 7, 14, and 21 after reperfusion (Supplementary Table 2, Fig. 3d and Supplementary Fig. 5c).

## Spatial distribution of cycling epithelial cells relative to IRI-induced necrotic cell death

To understand if the observed epithelial proliferative activity derived from random neighbor cells of necrotic cells or rather from scattered progenitor-like cells, we analyzed the spatial distribution of proliferating cells relative to necrotic cells detected at day 0. For this, we plotted the number of GFP-expressing tubule cells emerging in the immediate vicinity of a necrotic cell (radius < 25 µm), as well as the number of GFP-expressing cells emerging outside a radius of 25 µm of a given necrotic cell within a tubule segment. On average, we observed a significantly higher count of epithelial GFP cells located within immediate proximity of PI-positive nuclei (2.27 ± 0.15%) as compared to those detected in distant localization (1.62 ± 0.14%, mean ± SEM, $n = 336$ segments from 4 mice for both groups, $p < 0.001$, Supplementary Extended Statistics.l). Of note, closer evaluation demonstrated marked differences among segment types: In PT-S1 segments, cycling cells appeared almost exclusively adjacent to necrotic sites. In PT-S2 segments, cell cycling started in the close vicinity of necrotic sites. However, within the first 3 days, we detected a gradually growing population of GFP-expressing cells arising in distant localization of any necrotic cell detected at day 0. Similar dynamics were obtained for DCT/CD segments over time. However, GFP cells emerging at large distances from necrotic sites accounted for most of the proliferation observed in this group (Supplementary Table 3, Fig. 4a–d). This data indicated that while surviving neighboring cells accounted for a substantial part of proliferating cells,

additional inductive mechanisms seemed to be evident in PT-S2 and DCT/CD segments, respectively.

To better understand injury-dependent cell cycling in PT-S1 and PT-S2 segments, we analyzed dynamic GFP expression of non-necrotic (0% necrotic cell death at day 0), moderately necrotic (0–20% necrotic cells at day 0) and severely necrotic tubule segments (>20% necrotic cells at day 0). Consistent with the findings above, we observed significantly higher GFP expression in non-necrotic and moderately necrotic PT-S2 segments on days 2 and 3 as compared to respective PT-S1 (Fig. 4e). Among severely necrotic segments, PT-S1 segments proliferated more than PT-S2 segments on day 1. However, GFP expression in severely necrotic PT-S2 segments rapidly increased from day 2 and exceeded that observed in PT-S1 segments on days 3 and 7 (Fig. 4e). Finally, we aimed to evaluate if PT-S1 and PT-S2 displayed different capacities to proliferate upon necrotic cell death of a given degree. Thus, we performed a linear regression of PI-positive nuclei (% of total nuclei) observed on day 0 and the cumulative number of GFP-positive nuclei (% of total nuclei) observed on days 1 to 3 after partial IRI. For both PT-S1 and PT-S2 segments, a significant correlation between the initial number of necrotic cells and the subsequent proliferative response was detected (Fig. 4f). Of note, there was no significant difference in the slopes of the linear regressions, indicating a comparable necrosis-induced proliferative response in PT-S1 and PT-S2 segments (Fig. 4f). However, we detected a significantly higher intercept in the linear regression of PT-S2 segments (Fig. 4f), which demonstrated a population of PT-S2 segments that proliferated in absence of preceding necrotic injury. In contrast, the intercept of the PT-S1 linear regression equation was not statistically different from zero, confirming that proliferation in these segments strictly depended on the presence of initial necrotic injury.

## Granular casts precede proliferation in uninjured tubule epithelium downstream of necrotic injury sites

To understand why PT-S2 segments proliferated in the absence of necrotic cell death, we closely registered the observable remodeling processes in partial IRI kidneys over time. Necrotic injury in PT was associated with the shedding of dead tubule cells and apical membrane material into the tubule lumen (Figs. 1h, 4a, b). From day 1 after reperfusion, we observed pronounced luminal granular cast formation (Fig. 5c), which over time flowed downstream along the nephron (Fig. 4b, Supplementary Movie 1) and eventually also appeared in the lumen of previously non-necrotic nephron segments (Figs. 4b, 5a). We then quantified the luminal area of granular casts in the z-stack plane displaying the largest tubule diameter and normalized it to the cross-sectional area of the respective tubule segment at the same depth. Granular cast accumulation was detectable in necrotic (3.81 ± 0.51%, $n = 274$ segments from 3 mice) and non-necrotic PT-S2 segments (4.04 ± 0.43%, $n = 269$ segments from 4 mice) but was negligible in respective PT-S1 segments (necrotic: 0.14 ± 0.13%, $n = 221$ segments from 4 mice; non-necrotic: 0%, $n = 213$ segments from 4 mice), (Fig. 5d).

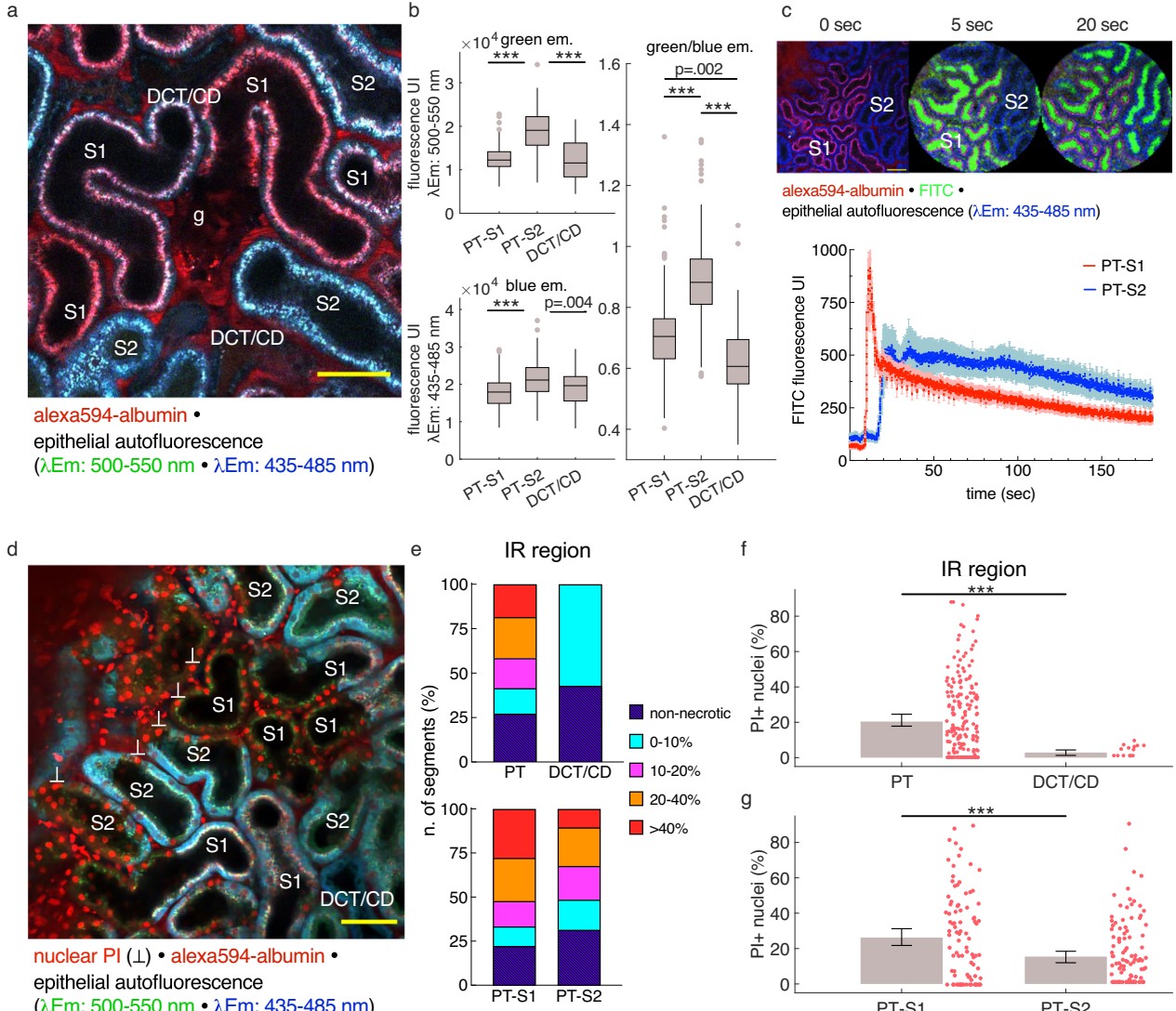

**Fig. 2 | Necrotic cell death distribution along the nephron in post-ischemic tissue. a** In vivo 2-photon image of a healthy mouse glomerulus (g) and surrounding tubules allows a clear distinction between proximal S1 (PT-S1), proximal S2 (PT-S2), and distal convoluted/collecting duct tubules (DCT/CD) based on distinct morphologic and autofluorescence patterns. Scale bar: 50 μm. **b** Tubule segment identity can be assigned based on quantitative assessment of green ($\lambda_{Em}$ = 500–550 nm) and blue ($\lambda_{Em}$ = 435–485 nm) tubular epithelial autofluorescence recorded at $\lambda_{Ex}$ = 750 nm (n = 152, 190, and 38 segments for PT-S1, PT-S2, and DCT/CD, from 4 mice 24h after partial IRI, respectively); boxplot, line at median, edges at 25th and 75th percentiles, whiskers from min to max, individual points considered outliers. Statistical test: linear mixed-effect model, p-values from two-sided tests (Supplementary Extended Statistics.f). **c** Upper panel: Representative in vivo 2-photon images acquired during bolus injection of FITC-4kDa-dextran reveal the arrival of filtered FITC-4kDa-dextran first in as PT-S1 and later in as PT-S2

classified tubules. Scale bar: 50 μm. Lower panel: Luminal FITC-4kDa-dextran fluorescence measured in PT-S1 and PT-S2 segments over time (n = 15 and 14 segments from 4 mice, respectively); mean ± SEM. **d** In vivo 2-photon image of a partial IRI kidney (ischemic region) at 2 h after reperfusion shows the distribution of propidium iodide (PI)+ nuclei across different tubule segments. Scale bar: 50 μm. **e** Necrotic injury distribution in IR regions of partial IRI kidneys across different nephron segments clustered by percentage of segments with indicated threshold of PI+ nuclei (% of total nuclei/segment). **f, g** Volumetric quantification of PI+ nuclei (% of total nuclei/segment) along different tubule segments in IR regions (n = 195, 90, 105 segments for PTs, PT-S1, PT-S2 from 6 mice, and 14 DCT/CD from 4 mice, respectively); mean ± 95% CI with scatterplot. Statistical test: linear mixed-effect model, p-values from two-sided tests (Supplementary Extended Statistics.g). *: $p < 0.05$; **: $p < 0.01$; ***: $p < 0.001$.

Of note, granular cast accumulation in PT-S2 segments spatially coincided with GFP expression one day later and seemed independent of initial necrotic injury in respective segments (Fig. 5a). To further explore an association between granular cast accumulation and cell cycling in PT-S2 segments, we plotted the extend of luminal granular casts over the percent of GFP-positive cells emerging outside a radius of 25 um of a given necrotic nuclei in the same tubule segments 24 h later. Linear regression analysis for non-necrotic PT-S2 segments demonstrated a significant correlation between the extent of granular cast accumulation and GFP expression observed one day later (Fig. 5e). In contrast, the same analysis done solely on necrotic PT-S2 segments

resulted in a significantly lower slope and R², but higher intercept (Fig. 5e). Taken together, these longitudinal analyses of remodeling events in partial IRI-kidneys indicated injury propagation along the proximal tubule, as determined by granular cast accumulation in previously non-necrotic downstream tubule segments, which correlated with their subsequent commitment to proliferate.

We next tested the hypothesis that delayed proliferation in granular cast-accumulating, non-necrotic tubule segments occurred independently of potentially sublethal IRI-induced injury. To achieve this, we inflicted selective necrotic injury in single PT-S1 segments to investigate if this would provoke granular cast accumulation and

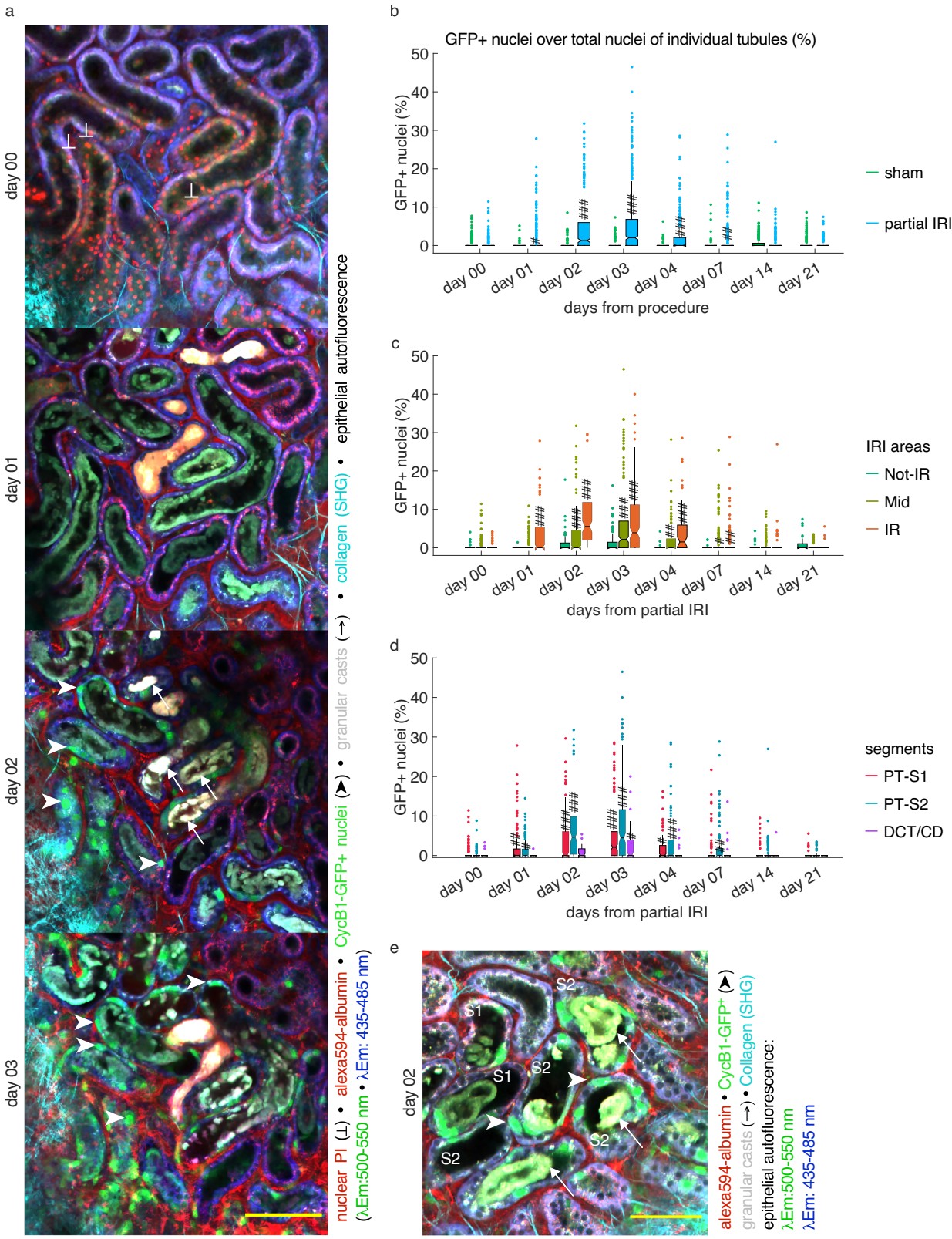

proliferation in downstream PT-S2 segments. Thus, we used the 2-photon laser as a micromanipulator[14,28–31] to induce selective and focal injury in PT-S1 segments of healthy CycB1-GFP kidneys through focused laser exposure. This resulted in selective PI uptake in the laser-targeted PT-S1 epithelium with no detectable necrotic injury in the surrounding renal interstitium or tubule epithelium (Fig. 6b, c). We then followed up upon potential remodeling processes over time using

serial 2PM. As expected, laser injury triggered pronounced epithelial proliferation in the targeted PT-S1 segments (Fig. 6b, c). Of note, focal laser-induced PT-S1 injury also induced granular cast accumulation in some of the surrounding PT-S2 tubule segments. Consistent with our previous findings, we observed significantly more GFP+ nuclei in PT-S2 segments that accumulated luminal granular casts than in those which did not (8.35 ± 1.47%, $n$ = 39 vs. 0.46 ± 0.07%, $n$ = 461, $p$ < 0.001, Fig. 6d,

**Fig. 3 | Dynamic epithelial proliferation in partial IRI kidneys. a** Serial in vivo 2-photon microscopy of an ischemic region in a partial IRI CycB1-GFP reporter mouse kidney on days 0, 1, 2, and 3 after reperfusion shows increasing GFP expression (arrowheads) over time. Scale bar: 100 μm. **b**–**d** Volumetric quantification of GFP expression (% of total nuclei/ segment) across different experimental groups (b: sham and partial IRI), different partial IRI injury regions (c: Not-IR, IR, and Mid), and different nephron segments located in IR and Mid regions of partial IRI kidneys (d: PT-S1, PT-S2 and DCT/CD). Data from n = 3 sham and 9 partial IRI mice. Detailed information on tubule segment number per group is provided in Supplementary Table 1 (b, c) and Supplementary Table 2 (d); boxplot, line at median, edges at 25th and 75th percentiles, whiskers from min to max, individual points considered outliers. An overview of the same dataset, including expanded statistical comparison, is shown in Supplementary Fig. 5. Statistical test: linear mixed-effect model, p-values from two-sided tests (Supplementary Extended Statistics.j). #: significant difference within group when compared to respective day 0; #: p < 0.05; ##: p < 0.01; ###: p < 0.001. **e** Representative in vivo 2-photon image of epithelial GFP expression in an IR region of a CycB1-GFP kidney at 2 days after partial IRI. Scale bar: 75 μm.

Supplementary Extended Statistics.u) and linear regression analysis demonstrated a significant correlation between the abundance of luminal granular casts in uninjured PT-S2 segments and evident GFP expression 24 h later (Fig. 6e). Furthermore, daily EdU injections during laser injury experiments followed by correlative imaging, confirmed actual proliferation of GFP-expressing nuclei as observed in vivo through clear overlap with EdU-incorporating nuclei as visualized ex vivo (Supplementary Fig. 4b). To test for unspecific laser-inflicted sublethal injury and proliferation in surrounding tubule segments, we analyzed the total number of epithelial GFP+ nuclei in different area zones defined by increasing distance to the basolateral membrane of the respective laser-targeted PT-S1 segments (Fig. 6f). In case granular cast accumulation and subsequent GFP expression in non-targeted PT-S2 segments were due to unspecific and sublethal laser injury, the number of GFP+ nuclei should be highest in immediate vicinity of the injury site. However, tubules directly adjacent to the laser-injured PT-S1 segments (0–35 μm) did not reveal more elevated epithelial GFP expression when compared to tubules distanced between 35–85 and 85–135 μm from the injured PT-S1 segment (Fig. 6f). This data suggested that granular cast-accumulating PT-S2 tubule segments accounted for downstream parts of the same nephron as the laser-targeted PT-S1 segment and proliferated independently of any unspecific laser injury. Consistent with this assumption, granular cast-accumulating and proliferating PT-S2 segments were often located at a reasonable distance from the laser-targeted PT-S1 segment (Fig. 6c), while most of the injury-surrounding tubule segments revealed no signs of injury and/or cellular remodeling (Fig. 6b, c). Taken together, these data indicate that selective laser-induced PT-S1 injury provokes granular cast accumulation in healthy downstream tubule epithelia, which thereafter commit to proliferation.

**Accumulation of granular casts associated with tubule atrophy**
Longitudinal intravital 2PM imaging of partial IRI kidneys over 21 days documented post-ischemic epithelial injury and subsequent remodeling events, which either lead to tubule atrophy or recovery (Fig. 7 and Supplementary Fig. 6). When analyzing the timely distribution of fate determination, we found that the fate of 77% of all recovering and 55% of all atrophic tubules was already determined by day 7 after reperfusion. By day 14, the fate of an additional 21% and 34% of recovering and atrophying tubules was determined (Fig. 7g), indicating that most fate-decisive remodeling processes took place within the first 1–2 weeks after reperfusion. To test if recovered tubules also regenerated functionally, we measured dynamic albumin reuptake capacity as the ratio of fluorescently labeled albumin in the apical tubule brush border and in the lumen of peritubular capillaries[14]. On day 0, the albumin reuptake capacity of recovering tubules was markedly reduced compared to uninjured partial IRI and sham PTs but recovered to sham levels by day 7. In contrast, albumin reuptake capacity in tubules which turned atrophic remained low over the entire observation period (Fig. 7a, b, Supplementary Movies 2 and 3).

When analyzing the distribution of atrophic, recovered, and uninjured tubule segments across the different partial IRI regions, we found that most atrophic tubules (68%) developed in the IR region, in which only 15% of the tubules recovered and 16% were uninjured. In Mid regions, 14% of the tubule segments turned atrophic, 47% recovered and 37% were uninjured. Not-IR regions consisted almost exclusively of uninjured tubules (97%) (Fig. 7f). Among necrotic tubule segments, atrophic tubules demonstrated significantly more necrosis on day 0 as compared to recovering tubules (23.78 ± 1.54%, n = 92 segments from 3 mice vs. 13.29 ± 1.04%, n = 89 segments from 4 mice, p < 0.001, Supplementary Extended Statistics.r), (Fig. 7c). Nevertheless, 30% of all atrophic tubules lacked any necrotic cell death on day 0. Interestingly, these tubules were typically located in close vicinity to strongly injured tubule segments and severely affected by granular cast accumulation (Supplementary Fig. 7), suggesting connection to necrotic tubule segments upstream. As demonstrated above, granular cast accumulation was also associated with the proliferation of initially non-necrotic tubule epithelium. However, high numbers of cycling cells did not predict tubule recovery. In contrast, we found significantly higher GFP expression (as assessed on day 3/4 after reperfusion) in tubules that turned atrophic as compared to recovering tubules (10.85 ± 0.94% vs. 4.92 ± 0.54%, n = 168 and 124 segments from 4 mice, respectively, p = 0.0065, Supplementary Extended Statistics.s). Therefore, we next tested if granular cast accumulation predicted tubule atrophy by plotting the luminal granular cast area (in % of total cross-sectional area) assessed at day 3/4 of a tubule segment and its probability of atrophy (0 = non-atrophy; 1 = atrophy) as determined at the end of the experiment. Binary regression analysis obtained a highly significant relationship (p < 0.001; $R^2$ = 0.81, Fig. 7h). Among tubules with severe and lumen-filling cases of granular cast accumulation, 87% turned atrophic, while only 13% recovered (n = 131 segments from 4 mice), indicating that granular casts dose-dependently predicted tubule atrophy. In contrast, tubules with flattened and simplified epithelium (indicative of ATI rather than ATN) mostly recovered (62%) and only 38% turned atrophic (n = 94 segments from n = 5 mice).

To investigate a possible role of atrophic tubules in AKI-CKD transition, we performed ex vivo immunohistology[32] for VCAM-1—a recently suggested marker of proximal tubule failed recovery[11] (n = 3). Thus, correlative imaging of the same immunostained kidney tissue as previously imaged in vivo (21 days after reperfusion) revealed epithelial VCAM-1 positivity selectively in atrophic tubules, while recovered tubules in the same kidneys remained VCAM-1 negative (Fig. 7e).

## Discussion
Serial intravital microscopy of the kidney using the AIW technique is a powerful approach to study structural and functional changes in the remodeling kidney over time and is based on longitudinal observations[14,29,32,33]. In this study, we exploited serial intravital 2PM of the living mouse kidney to investigate epithelial injury and recovery in a model of AKI, which was conceptualized to understand the effects of focal injury on uninjured renal tissue. First, we report significant differences in the susceptibility of PT-S1, PT-S2, and DCT/CD nephron segments toward post-ischemic necrotic injury. Second, we demonstrate that necrotic epithelial cells are largely replaced by the proliferation of surviving neighbor cells. Third, we document how post-ischemic necrotic cell loss and luminal cellular debris accumulation (granular casts) perpetuate renal injury and initiate epithelial

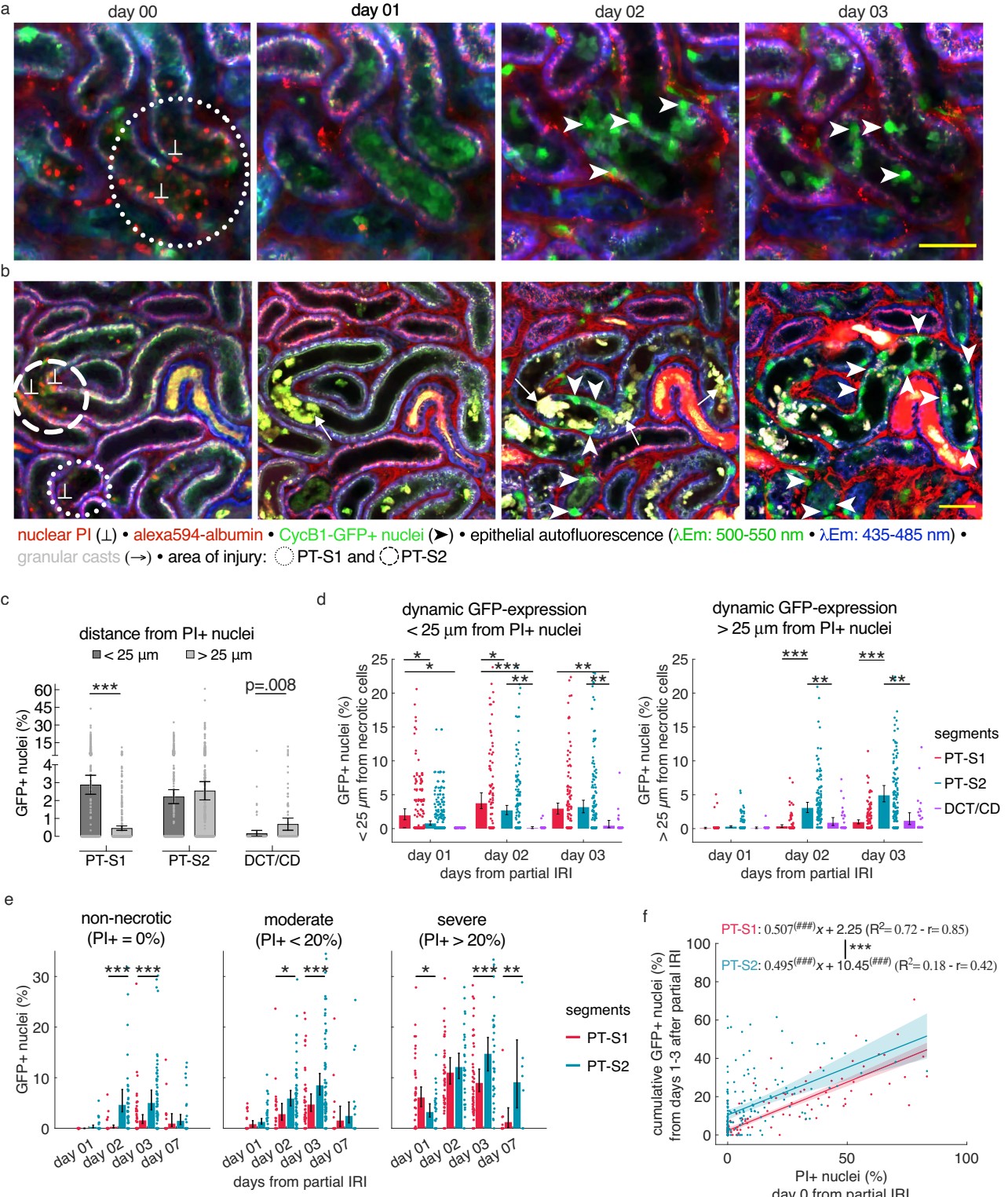

nuclear PI (⊥) • alexa594-albumin • CycB1-GFP+ nuclei (➤) • epithelial autofluorescence (λEm: 500-550 nm • λEm: 435-485 nm) • granular casts (→) • area of injury: ⊙ PT-S1 and ◯ PT-S2

remodeling and proliferation in downstream, previously non-necrotic tubule segments. Fourth, our data identify high post-ischemic necrotic injury and accumulation of granular casts as strong predictors of tubule atrophy development and demonstrate de novo expression of the pro-inflammatory and pro-fibrotic protein VCAM-1 in atrophic tubules, suggesting an active role of this tubule population in AKI-CKD transition.

In agreement with previous work[13,34], our study identified the proximal tubule as the main site of post-ischemic necrotic injury. While

it is generally accepted in the field that the proximal straight tubule (PT-S3) sustains the highest degree of post-ischemic injury, the majority of available data does not distinguish between PT-S1 and PT-S2[34,35], which are functionally considered alike and hard to separate from each other[34]. We and others[25,26,36] show that PT-S1 and PT-S2 can be reliably distinguished from one another using 2PM. In contrast to traditional research techniques, our serial imaging approach further allowed us to largely overcome the difficulties of classifying injured proximal tubule segments, as multiple time points were considered for

**Fig. 4 | Spatial distribution of proliferating cells relative to necrotic sites. a, b** Serial in vivo 2-photon microscopy images of representative Mid regions in CycB1-GFP reporter kidneys at indicated days after partial IRI. Note localized GFP expression (arrowheads) around necrotic sites (⊥) in PT-S1 segments while GFP expression arises also in remote distance of necrotic sites (⊥) in PT-S2. Scale bar: 50 µm. **c** Cumulative volumetric quantification of GFP-expressing nuclei (% of total nuclei/ segment) determined from days 1–3 after partial IRI for PT-S1, PT-S2, and DCT/CD, which arose either in proximity (<25 µm) or distant localization (>25 µm) from any propidium iodide (PI)+ cell detected on day 0 (n = 145, 184 and 37 segments from 4 mice for PT-S1, PT-S2, and DCT/CD, respectively); mean ± 95% CI with scatterplot. Statistical test: two-sided paired t-test with p < 0.05 considered significant (Supplementary Extended Statistics.k). **d** Same dataset as in (c) displayed to reveal distinct locations of proliferating tubule cells in different tubule segments over days after partial IRI (n = 4 partial IRI mice; detailed information on segment number/group provided in Supplementary Table 3); mean ± 95% CI with scatterplot. Statistical test: linear mixed-effect model, p-values from two-sided test (Supplementary Extended Statistics.m). **e** Volumetric quantification of tubular GFP expression (% of total nuclei/ segment) clustered by necrotic thresholds (n = 223, 140, 157 PT-S1 segments and 250, 237, 103 PT-S2 segments for non-necrotic, moderate, and severe, respectively, from 6 mice); mean ± 95% CI with scatterplots. Statistical test: linear mixed-effect model, p-values from two-sided test (Supplementary Extended Statistics.n). **f** Linear regression of PI+ nuclei (% of total nuclei/segment) at day 0 and cumulative GFP+ nuclei (% of total nuclei/ segment) from days 1, 2, and 3 for PT-S1 and PT-S2 segments (n = 144 and 180 from 4 mice). Fit (slope*x + intercept, plotted with 95% CI), $R^2$ and Pearson's coefficient (r) are reported for both. #: significant difference of slope and intercept from zero. *: significance test across groups, p-values from two-sided test. */#: p < 0.05; **/##: p < 0.01; ***/###: p < 0.001.

segment classification. Our data clearly demonstrate a higher susceptibility for necrotic injury in post-ischemic PT-S1 segments as compared to PT-S2. While PTs mainly rely on aerobic metabolism to meet their energy needs, anaerobic glycolysis may be critical in preserving viability in situations of reduced oxygen availability[37]. Consistent with our findings, inhibition of mitochondrial respiration in isolated perfused rat kidneys demonstrated a clear hierarchy of PT segment injury, with PT-S1 being the most susceptible, followed by PT-S2 and least PT-S3[38]. Due to insufficient laser penetration in renal tissue[25,39], deep-located PT-S3 segments could not be investigated in our study. The increased post-ischemic necrosis of PT-S1 as compared to PT-S2 could be explained by a lower capacity of PT-S1 to generate ATP from glycolysis. Consistent with such a hypothesis, previous studies demonstrated a higher vulnerability of PT-S2 toward inhibition of glucose metabolism than PT-S1[25,38].

Longitudinal imaging and analysis of epithelial injury and remodeling in CycB1-GFP reporter mice demonstrated a close spatial relationship between necrotic and cycling cells. Within 24 h of post-ischemic necrotic injury, surviving cells in close vicinity to necrotic sites committed to proliferation. Consistent with a higher degree of initial necrotic injury in PT-S1, cell proliferation at early time points (24 h after partial IRI) was highest in PT-S1. However, the proliferative activity in downstream nephron segments drastically increased over time. Thus, the correlation of spatial and temporal information from the same renal cells over several days ultimately revealed that proliferation in PT-S1 segments was mostly restricted to neighboring cells of necrotic sites, while PT-S2 and DCT/CD segments demonstrated a growing population of cycling cells in remote distance of initial necrotic injury. Strikingly, we observed that this phenomenon spatially and temporally correlated with preceding granular cast accumulation in the lumen of respective tubules. Consecutive imaging of the same tubule segments over several days clearly demonstrated the appearance of granular casts at sites of necrotic injury, which slowly traveled downstream into previously non-necrotic tubule segments and coincided with epithelial GFP expression along the way (Fig. 4b). Using an additional selective PT-S1 injury model, we further demonstrated that this phenomenon also occurred in completely healthy PT-S2 segments downstream of the injury site and in the absence of any potentially sublethal effects of ischemia on non-necrotic tubules. Of note, accelerated epithelial proliferation was not associated with successful recovery (Supplementary Movie 1). On the contrary, we detected a higher number of cycling cells in tubules, which eventually turned atrophic, than in those which recovered. The increased likelihood of tubule atrophy in the presence of enhanced epithelial proliferative activity may be surprising. Nevertheless, our data indicated a linear relationship between the initial degree of tubular necrotic cell death and the number of proliferating tubule cells arising thereafter. Thus, the increased proliferative activity observed in atrophying tubules was also preceded by a higher degree of epithelial necrosis and the formation of granular casts. Consistent with this, granular cast

accumulation did not only associate with progressive epithelial proliferation but predicted tubule atrophy development in a dose-dependent fashion. Overall, these observations indicate a previously underestimated role of early proximal tubule necrosis in AKI and demonstrate evidence for substantial and fate-determining injury propagation along the nephron. Even though these processes still initiated epithelial repair programs, the latter was eventually deemed unsuccessful in the presence of too severe epithelial necrosis. Consistent with a concept of necrosis-induced injury propagation into downstream nephron segments, old light and electron microscopy data from post-ischemic rat kidneys illustrated necrosis in proximal convoluted tubules, while proximal straight tubule segments were entirely blocked by accumulated luminal blebs[35]. Furthermore, membrane blebbing and granular cast formation during AKI have been associated with tubule obstruction and backflow[34,35,40], which may favor tubule atrophy. Lastly, membrane rupture during necrotic cell death forms cellular debris and leads to the extracellularization of so-called danger-associated molecular patterns (DAMPs), which may further drive tissue inflammation and injury, a process commonly referred to as necroinflammation[41].

Of note, we also observed how moderately necrotic PT-S1 segments presented later without classic signs of ATN (e.g., granular cast formation) but rather displayed a phenotype of ATI, with simplified and thinned epithelium, which lacked brush border lining (Fig. 4a day 2–3 and Fig. 7a day 3 Mid). Since AKI patient biopsies often rather display features of ATI, the role of ATN in human AKI is controversially discussed[22,42]. Our longitudinal approach allowed following the same renal tubules over time and demonstrated that ATI can be preceded by mild to moderate tubule necrosis. Biopsies from AKI patients are not routinely collected, are limited in number, and are taken at heterogeneous time points after the insult[22], making it difficult to exclude the role of necrotic cell death in human AKI. Furthermore, urinalysis from AKI patients demonstrated muddy brown casts with numerous renal epithelial cells, suggestive of ATN[43]. These observations indicate the need for more patient material to re-assess the role of ATN in human AKI.

Failed tubule repair may be a driver of AKI-CKD transition. Using correlative ex vivo histology, we clearly demonstrated tubular de novo expression of VCAM-1 selectively in atrophic tubules. Unlike common tubule injury markers, such as Kim1 or NGAL, VCAM-1 expression does not occur during the acute injury phase but starts rising between day 3 and 7 after reperfusion[8,11], which is also when tubule atrophy was first observed in our data set (Fig. 7g). VCAM-1 is a recently established marker of a pro-inflammatory proximal tubule cell state that engages in immune cell activation and profibrotic signaling[11]. Consistent with our findings, a recent study linked renal VCAM-1 expression with progressing kidney atrophy, as determined by decreased kidney weight after AKI[8]. Thus, our results indicate that tubule atrophy may present more than an inactive tubule state but pinpoint an active role of this population in AKI-CKD transition.

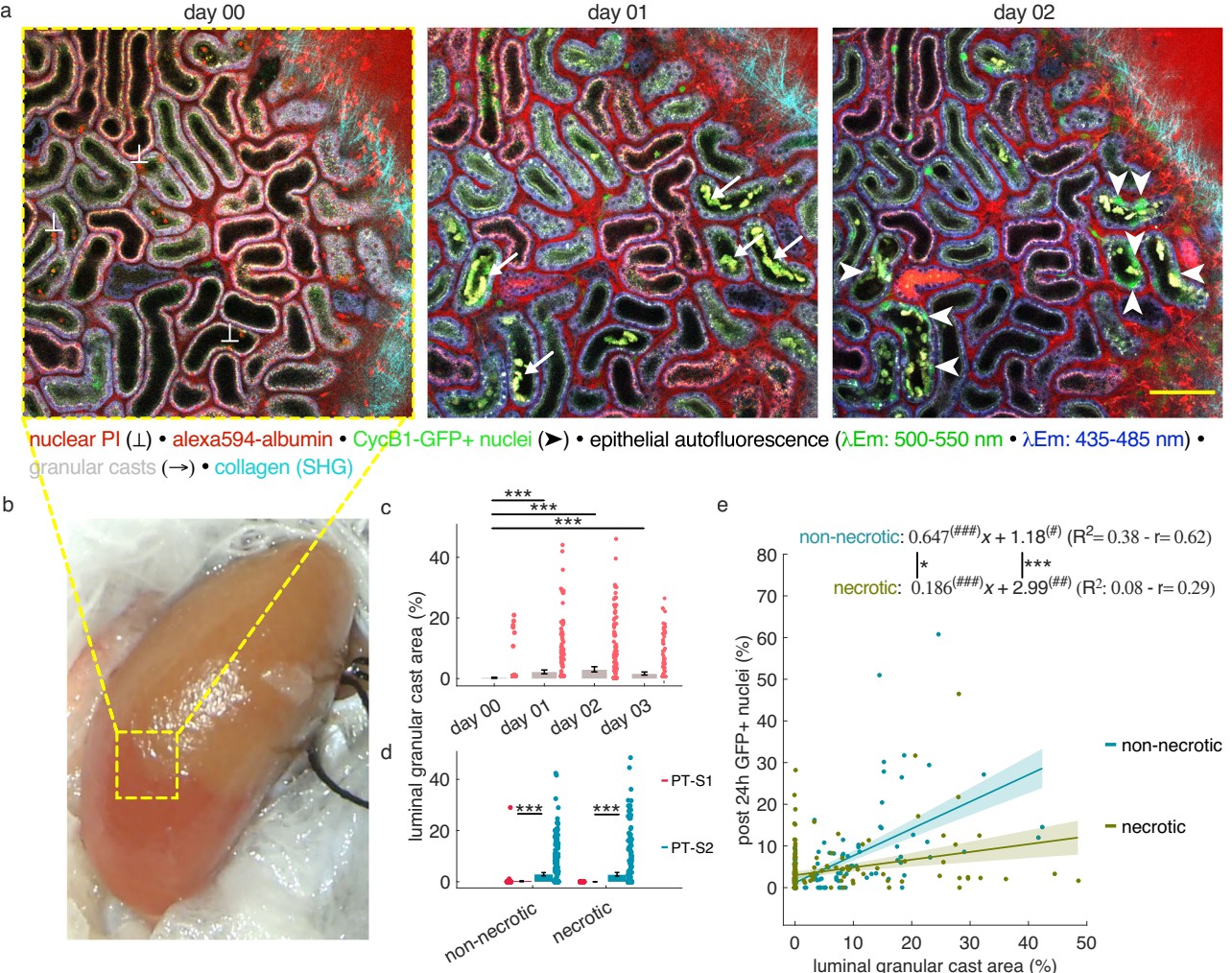

**Fig. 5 | Granular cast accumulation precedes proliferation in non-necrotic tubule epithelium downstream of necrotic injury sites. a** Representative serial in vivo 2-photon microscopy images of a Mid region in CycB1-GFP kidneys at day 0, 1, and 2 after partial IRI. Note how GFP+ cells (arrowheads) appear 1 day after granular cast accumulation (arrows) in initially non-necrotic segments (PI-negative at day 0). Scale bar: 100 μm. **b** Representative stereomicroscope photo of a mouse kidney during renal artery branch occlusion showing a typical border region between the perfused and ischemic kidney regions. **c**, **d** Luminal granular cast area measured at the largest cross-sectional plane of each tubular segment and normalized by the cross-sectional area of the respective segment at the same imaging depth, grouped by days after partial IRI (c) (n = 468, 328, 329, and 320 segments for respective days from 6 (day 00) and 4 (day 01-03) mice), and segment type in the absence or presence of necrosis (d), respectively (n = 213 and 221 for non-necrotic and necrotic PT-S1 from 4 mice; 269 and 274 segments for non-necrotic and necrotic PT-S2 from 4 and 3 mice, respectively). Any tubule with >1% PI+ necrotic cells at day 0 was classified as necrotic; mean ± 95% CI with scatterplots. Statistical test: linear mixed-effect model, p-values from two-sided test, no adjustment for multiple comparisons (Supplementary Extended Statistics.o). **e** Linear regression of luminal granular cast area (% of total segment cross-sectional area) and GFP+ nuclei (% of total nuclei/ segment, assessed in 3D) measured 24 h later in the same segment and proliferating outside a 25 μm radius from any PI+ nucleus (n = 179 and 184 for non-necrotic and necrotic segments from 4 and 3 mice, respectively). Fit (slope*x + intercept, plotted with 95% CI); R² and Pearson's correlation coefficient (r) are reported for both. #: significant difference of slope and intercept from zero. *: significant difference between the segment types, p-values from two-sided test, *$/#: p < 0.05; **/##: p < 0.01; ***/###: p < 0.001.

In this study, we display important insights into how tubule injury links to tubule remodeling and fate. Our data demonstrate a key role of early proximal tubule necrosis and subsequent injury propagation for tubule fate and atrophy development. Taken together, these findings suggest a potential therapeutic intervention window within the first days after AKI to limit injury propagation and tubule atrophy.

## Methods
### Experimental animals

All experimental procedures involving animals in this study were approved by local authorities (Animal Experiments Inspectorate, Denmark, permit number: 2020-15-0201-00443) and reported according to ARRIVE guidelines. Tg(Pgk1-Ccnb1/EGFP)1Aklo (CycB1-GFP) reporter mouse line was purchased by The Jackson Laboratory

(Strain #:023345) and bred in the barrier animal facilities at the Department of Biomedicine, Aarhus University. Littermates were weaned at 3 weeks of age. Adult mice were housed at the Preclinical Research Facility at Aarhus University Hospital and at the Animal Facility at the Department of Biomedicine, Aarhus University. Mice were kept on a 12-h light: dark cycle with ad libitum access to a standard diet (#1324, Altromin, Germany) and water in 2–5 housing groups in individually ventilated cages (Technoplast model GM500 or GM9000) at 21 ± 2 °C and 55 ± 10% relative humidity. Bedding consisted of Aspen wood size 2–3 mm (bedding midi, Abedd, Latvia) and compressed cotton tubes for nesting, and home cages were enriched with Mouse Igloo with spinning wheels, wooden chew sticks, cardboard tubes, and Shepherd Shacks. Cages were routinely cleaned once a week.

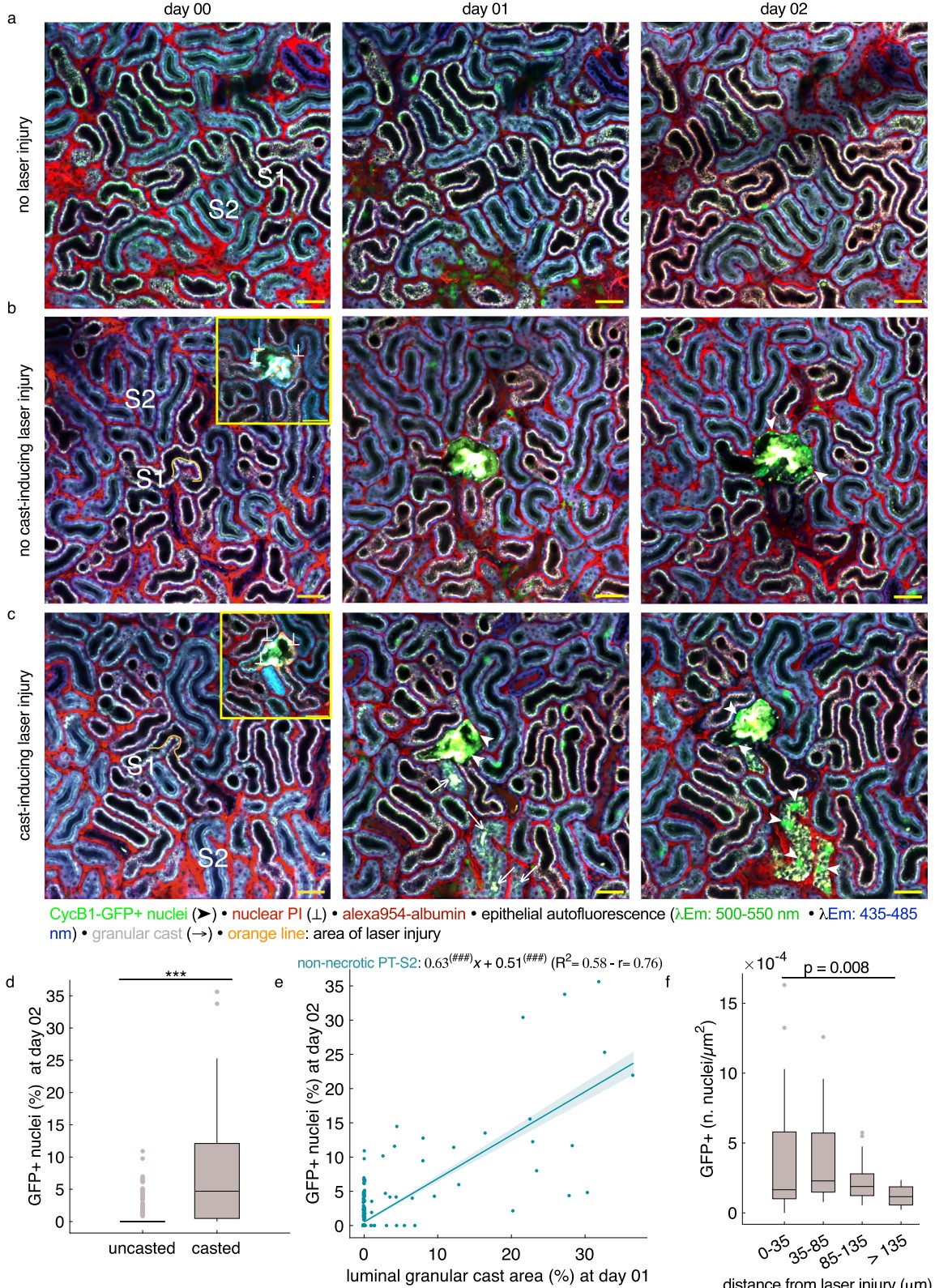

CycB1-GFP+ nuclei (➤) • nuclear PI (⊥) • alexa954-albumin • epithelial autofluorescence (λEm: 500-550 nm • λEm: 435-485 nm) • granular cast (→) • orange line: area of laser injury

non-necrotic PT-S2: $0.63^{(\#\#\#)}x + 0.51^{(\#\#\#)}$ ($R^2$= 0.58 - r= 0.76)

## Experimental groups

Males and female mice with a mean weight of $26.9 \pm 0.8$ g and age of $17.7 \pm 2.3$ weeks of age (mean $\pm$ SEM) were used for partial IRI in vivo experiments. The study design is summarized in Supplementary Table 4. Sex comparison was not considered in the original study design, and n-numbers do not allow final conclusions. As AKI is more severe in male subjects[44], we mainly collected data from male suspects

($n = 21$ males and $n = 5$ females). Mice were randomly allocated to the following treatment: (1) partial ischemia-reperfusion injury (partial IRI) and (2) sham control surgeries. For both of the two groups above, two experimental protocols were performed on separate subjects: serial intravital imaging (11 partial IR, 3 sham) and GFR measurement in freely moving mice (8 partial IRI, 4 sham). Within the partial IR imaging group, $n = 6$ mice were assigned to short-term serial imaging protocol

**Fig. 6 | Luminal granular cast accumulation and proliferation in uninjured PT-S2 segments after selective focal PT-S1 injury.** Serial in vivo 2-photon microscopy images of representative CycB1-GFP kidney regions without (**a**) and with selective focal PT-S1 injury (**b**, **c**) through targeted 2-photon laser exposure (orange line). **b** Representative image series before and after laser-induced focal PT-S1 injury in which no granular cast accumulation was detectable in surrounding PT-S2 segments. Note local proliferation (arrowheads) in targeted PT-S1 segment with lacking unspecific epithelial injury or proliferation in surrounding tubule segments. **c** Representative image series before and after laser-induced focal PT-S1 injury in which luminal granular cast accumulation (arrows) and subsequent GFP expression (arrowheads) was detectable in surrounding, initially uninjured PT-S2 segments of remote distance to the injury site. Scale bar: 50 μm. **d** Volumetric quantification of GFP+ nuclei (% of total nuclei/segment) in non-granular cast-accumulating (uncasted) and granular cast-accumulating (casted) PT-S2 segments at day 02 after focal PT-S1 injury ($n = 461$ and 39 for uncasted and casted segments, from 7 mice); boxplot, line at median, edges at 25th and 75th percentiles, whiskers from min to max, points considered outliers. Statistical test: linear mixed-effect model, p-values from two-sided test (Supplementary Extended Statistics.u). **e** Linear regression of luminal granular cast area (% of total segment cross-sectional area) at day 01 and GFP+ nuclei (% of total nuclei/segment) measured 24 h later in uninjured PT-S2 segments ($n = 498$ from 7 mice). Fit (slope*x + intercept, plotted with 95% CI); $R^2$, and Pearson's coefficient (r) are reported. #: significant difference of slope and intercept from zero. **f** Normalized number of GFP+ nuclei detected in different area zones defined by increasing distance from the basolateral membrane of the laser-injured PT-S1 segment indicates no unspecific laser-induced proliferation in injury-adjacent tubules (0–35 μm). ($n = 20$ for each area zone from 7 mice); boxplot, line at median, edges at 25th and 75th percentiles, whiskers from min to max, individual points considered outliers. Statistical test: linear mixed-effect model, p-values from two-sided test (Supplementary Extended Statistics.u). */#: $p < 0.05$; **/##: $p < 0.01$; ***/###: $p < 0.001$.

(image acquisition at day 0, 1, 2, 3 after partial IR), and $n = 5$ mice to long-term imaging protocol (image acquisition at day 0, 3, 4, 7, 14, 21 after partial IR). Partial IRI mice were included in the study upon visual confirmation of temporal restriction and subsequent reinstalment of blood flow in approximately half of the kidney tissue (as shown in Fig. 1a, d). Partial IRI mice were excluded from the study if the incorrect placement of the abdominal imaging window did not allow identification of Mid or IR regions using intravital microscopy.

Blinding of experimental treatment could not be performed in our design as the validation of the expected surgical outcome required close microscopic examination during surgery.

For confirmation of reliable PT-S1 and PT-S2 classification, four 10-week-old male C57/Bl6 mice were used. For experiments applying selective laser-induced PT-S1 injury, seven $8.7 \pm 1.1$ weeks (mean ± SEM) old male CycB1-GFP mice were used.

### Surgical intervention: partial IRI and AIW implantation

The surgical preparations consisted of two procedures: induction of AKI via a partial IRI model developed for this study and implantation of an abdominal imaging window (AIW) placed simultaneously over both injured and uninjured sides of the kidney (Fig. 1d) for serial intravital microscopy. The AIW implant was prepared by gluing a 12 mm coverslip with cyanoacrylate glue on the upper side of an implant made of two thin titanium rings (14 mm in diameter), separated by a 1.5 mm width groove[32,45–47].

Mice were anesthetized with isoflurane (3.5% for induction, 1.5–1.75% for maintenance, 1.2–1.8 L/min flow rate, 50% oxygen in medical air), received buprenorphine (0.1 mg/ kg BW, Temgesic) via i.p. injection and were placed on a $37 \pm 0.5\,°C$ heating pad with rectal thermometer (hB101/2, Harvard apparatus, Cambridge, MA, US). Eye ointment prevented cornea dehydration. The mouse's left flank was shaved and disinfected with Chlorhexidine (0.5% in 70% Ethanol), and a 1 cm mm dorsoventral incision was made at the midline between the ribcage and the left knee. The left kidney was gently repositioned via maneuvering of connective tissue so its ventral surface was slotted at the center of the incision. After careful dissection of connective tissue, the arterial bifurcation at the pedicle was exposed (Fig. 1a, d), and the right artery branch was occluded with a 6-0 silk thread for 21 min while temperature and hydration of the kidney were maintained with a Spongostan sponge (Ethicon) continuously flushed with $36.5\,°C$ saline. After reperfusion, the abdominal wall surrounding the kidney was sutured to the skin by a purse-string suture for abdominal imaging window (AIW) implantation. Using 10 μl of cyanoacrylate glue, the kidney was glued to the AIW implant. Careful placement of the AIW above the kidney ensured that post-ischemic and non-ischemic tissue areas would be accessible (Fig. 1d). Finally, the skin was secured around the groove of the AIW by tightening the purse-string suture[32,46]. Then, 10 μl/g BW sterile saline was injected i.p. for fluid supplementation.

Post-operative analgesia consisted of 1–3 mg/kg BW Meloxicam s.c. administered after surgery and equal doses of Meloxicam and Temgesic once per day for the 2 days following the surgery. Mice recovered in a heated chamber for 2 h before the first imaging session.

### Glomerular filtration rate and albumin/creatinine measurements

To characterize the impact of partial IRI on renal function, we measured glomerular filtration rate (GFR) transcutaneously over 7 weeks in freely moving sham ($n = 4$) and partial IRI mice ($n = 8$) using transcutaneous GFR monitoring (Medibeacon, Germany)[48]. In short, mice were anesthetized with isoflurane (3,5% induction dose) and thoroughly shaved using depilatory cream. Then, a miniaturized fluorescence detector (Medibeacon, Germany) was taped to the ribs of the animal, and a FITC-conjugated inulin-analog, sinistrin, was administered by retro-orbital injection (5 mg/100 g BW; Medibeacon). Isoflurane was discontinued and the mouse was allowed to wake up. Measurement lasted for 90 min. GFR was analyzed as the clearance rate of FITC-sinistrin following the instruction manual using the 3-compartment model.

Albuminuria was assessed as albumin/creatinine ratio in spot urine samples[49,50] and over a time course of 4 weeks ($n = 3$ sham and $n = 4$ partial IRI). Following collection, urine was frozen at $-80\,°C$ until analysis. Urinary albumin and creatinine content was analyzed using commercially available mouse albumin ELISA kit (Nordic Biosite, Product. NO. E99-134) and Creatinine Urinary detection kit (Thermo Fisher, Product. NO. AIACUN), respectively.

### Serial intravital 2-photon microscopy

We performed serial intravital microscopy for up to 3 weeks to track the dynamics of individual tubular epithelial cells over time in sham and partial IRI mice. In vivo 2PM was performed using a Chamelion Ultra II Coherent Laser and an Ultima Investigator Plus multiphoton setup (Bruker Corporation, Billerica, MA, USA) with PraireView IV software in upright objective configuration (20X Olympus XLUMPLFLN Objective, water immersion, 1.00 NA, 2.0 mm WD). Excitation wavelengths used were 750 nm and 940 nm. Emission signal was detected after a 720sp filter on 3 GaAsP (Hamamatsu, H7422-40) photomultipliers (Ch1: $\lambda_{et} = 595/50$ nm, Ch2: $\lambda_{et} = 525/50$ nm, Ch3: $\lambda_{et} = 460/50$ nm). In addition, we used an upright Olympus FVMPE-RS 2-photon microscope (Olympus, Japan) with Fluoview FV31S software (Olympus, Japan) that was equipped with a MaiTai HP DS-OL excitation laser (Spectra Physics, United States), XLPLN25xWMP2 objective, water immersion, (Olympus, Japan, NA 1.05; WD 2.00 mm), and the following detection cubes: Ch1: $\lambda_{et} = 610/70$ nm (multialkali PMT), Ch2: $\lambda_{et} = 540/40$ (GaAsP), Ch3: $\lambda_{et} = 480/40$ (GaAsP).

Prior to imaging, mice were anesthetized with isoflurane (same induction dose of surgery) and 10 μl of propidium iodide (PI,

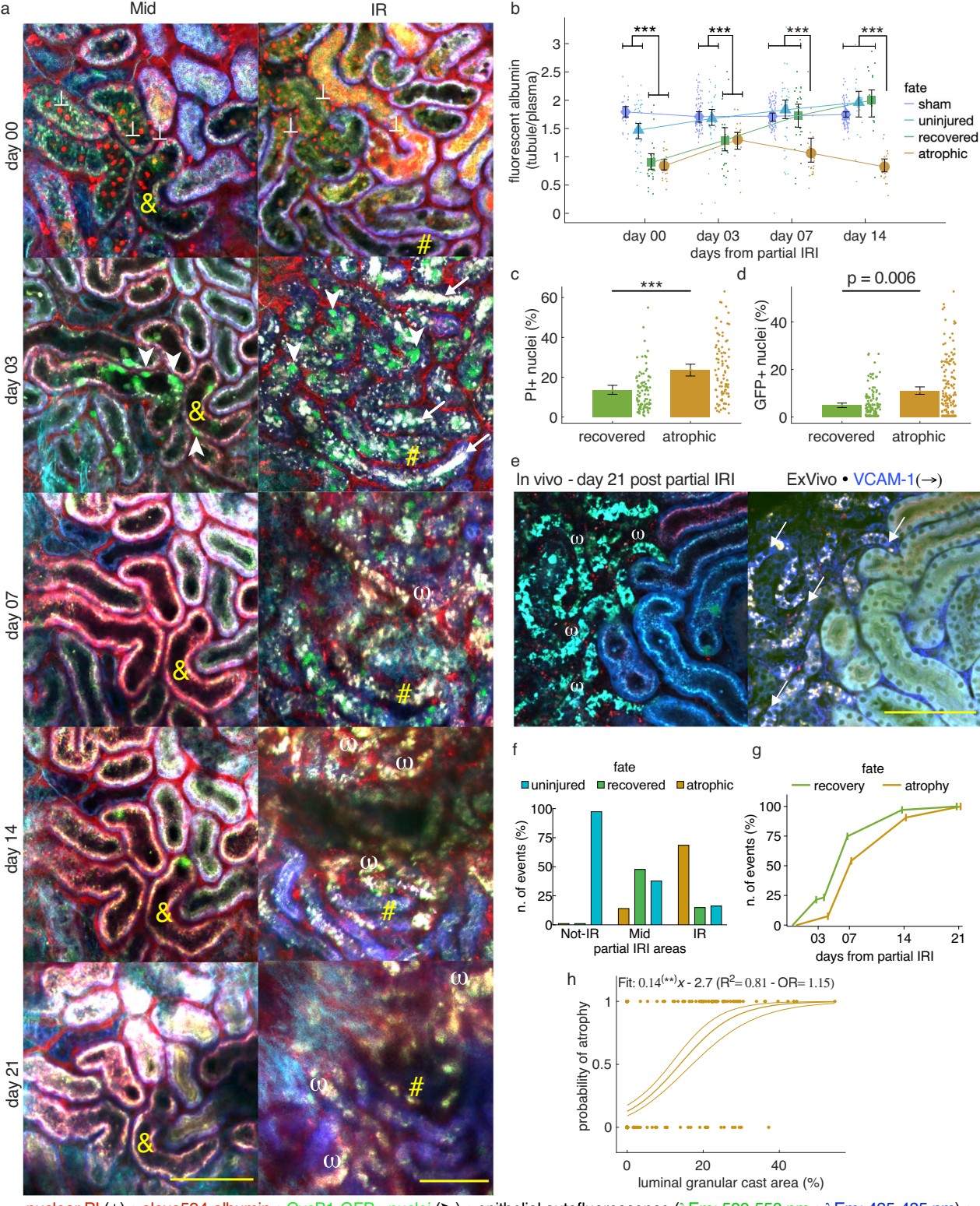

nuclear PI (⊥) • alexa594-albumin • CycB1-GFP+ nuclei (➤) • epithelial autofluorescence (λEm: 500-550 nm • λEm: 435-485 nm) •
granular casts (→) • collagen (SHG) • atrophic tubuli (ω) • placeholder over serial imaging (#,&)

0.25 mg/ml, Thermo Fisher) and 1 μl/g BW of a 2.5 mg/ml conjugated Albumin-Alexa Fluor 594 dye solution (Alexa594-albumin, Invitrogen) were injected retro-orbitally. PI is a charged DNA-intercalating dye, which specifically penetrates necrotic cells due to impaired membrane integrity[51]. For confirmation of PT-S1/PT-S2 segment classification, 4 kDa dextran conjugated to FITC was injected as a bolus through the tail vein (10 μl/bolus of 40 mg/ml solution in PBS).

For serial imaging, the AIW implant was slotted into a custom 3D-printed imaging frame providing image stabilization on an upright microscope setup[52]. During imaging, anesthesia and temperature were maintained with a low-flow (30–50 ml/min, 1.2–1.5%) isoflurane vaporizer (SomnoSuite, Kent Scientific), which was equipped with a heating blanket placed below the animal.

**Fig. 7 | Tubule remodeling processes in relation to tubule fate and functional recovery. a** Serial in vivo 2-photon microscopy images of representative Mid and IR regions in CycB1-GFP kidneys at indicated days after partial IRI show GFP expression (arrowheads) in flattened tubule epithelium (FL) and granular casts (arrows) in atrophic tubules (ω). Scale bar: 100 μm. **b** Dynamic assessment of PT-S1 albumin reuptake over time ($n = 80, 40, 28$, and $22$ for sham, uninjured, recovered, and atrophic tubule segments from 3 sham and 4 partial IRI mice); mean ± 95% CI with scatterplot. Statistical test: linear mixed-effect model, *p*-values from two-sided test (Supplementary Extended Statistics.p). **c, d** Volumetric quantification of PI+ nuclei (% of total nuclei/segment) at day 0 (from tubules with >1% PI+) (**c**) and GFP+ nuclei (% of total nuclei/ segment) at day 3/4 (**d**), in recovered and atrophic tubule segments, respectively (**c**: $n = 89$ and $92$ segments from 4 and 3 mice; **d**: $n = 124$ and $168$ segments from 4 mice, respectively); mean ± 95% CI with scatterplot. Statistical

tests: linear mixed-effect model, *p*-values from two-sided test (Supplementary Extended Statistics.q). **e** Correlative in vivo and ex vivo 2-photon microscopy images of CycB1-GFP kidneys 21 days after partial IRI reveal selective tubular VCAM-1 immunostaining (blue, arrows) in atrophic tubules (ω). Scale bar: 100 μm. **f, g** Tubular fate distribution (%) across partial IRI areas (**f**: $n = 82,232$, and $153$ segments for Not-IR, Mid, and IR from 4, 6, and 5 mice, respectively), and days after partial-IRI (**g**: $n = 103$ and $106$ for recovered and atrophic tubules from 6 mice, respectively). **h** Binary regression of luminal granular cast area (in % of total segment cross-sectional area) at day 3/4 and respective tubule fate (0 = non-atrophy; 1 = atrophy), ($n = 237$ segments from 4 mice, respectively). Fit (slope*x + intercept); $R^2$, and effect size as odds ratio (OR) for the predictor are reported. *: $p < 0.05$; **: $p < 0.01$; ***: $p < 0.001$.

## Acquisition protocol

First, we acquired a complete atlas of the AIW-attached kidney surface at 750 nm excitation wavelength, which allowed tissue navigation during consecutive imaging sessions. We identified areas of intense necrosis by a combination of nuclear-specific PI signal and a visible reduction of tubular epithelial autofluorescence emitted in the blue spectrum at 750 nm excitation wavelength[25]. In mice undergoing partial IRI, we identified three reproducible regions (Fig. 1e): (1) ischemic reperfusion damaged tissue (IR regions); (2) border regions surrounding the ischemic side with a limited amount of necrosis (Mid regions); (3) uninjured tissue, Not-IR region. In sham mice, all tissue was classified as a Not-IR region.

Next, we pinpointed 4–8 field-of-views (FOV) over the atlas across the three region types. We imaged each FOV with dual-excitation wavelengths over three emission channels (described above) using the following acquisition sequence: (1) 940 nm excitation, single image at 20–30 μm deep from kidney capsule, defined as the midpoint section of FOV (1024 × 1024, dwell time: 3.2 μs, Galvo aster scan); (2) 940 nm excitation, 60 μm deep volumetric series of range −30 to +30 μm from FOV midpoint (512 × 512, dwell time: 2.4 μs, Galvo raster scan, 1 μm step size); (3) and (4), identical parameters of (1) and (2) after changing excitation wavelength to 750 nm; (5) motorized return at midpoint section of FOV to assess potential misplacement of individual tubular cells during acquisition. Laser intensity was calibrated to deliver a range of 20–35 mW laser power at the objective. Zoom factor used ranged from 1 to 2.5X. The acquisition sequence was embedded as PrareView Scripts function to minimize user error.

For confirmation of PT-S1/PT-S2 segment classification, a time lapse of 13.7 Hz was recorded during bolus i.v. injection of 4 kDa dextran conjugated to FITC.

For selective injury of PT-S1 segments, we used the 2-photon laser as a micromanipulator[14,28–31] to selectively ablate defined cells in a single PT-S1 segment within a FOV. Thus, we specifically targeted approximately 35 adjacent PT-S1 tubule cells through repeated line scans (pixel dwell time: 4 μs/pixel) over the selected tubule epithelium using 750 nm excitation. Subsequently, mice were imaged daily for the following 3 days to investigate remodeling processes within the injured and the surrounding uninjured tubule segments. Control FOVs at remote distance to the laser injury were acquired alongside. Prior to each image session, mice were administered PI and Alexa594-albumin, as described above.

## Imaging analysis

We performed manual image analysis and cell count on volumetric scans using FIJI[53] Bio-Formats and BIOP Multi-Stack-Montage plugins[54]. The two excitation tracks for each image sequence were merged into a single composite image (Supplementary Fig. 3) using BIOP-Multi Stack Montage. Assignment of pseudocolors in merged images was done to grant proper display of merged images as described in detail in Supplementary Fig. 3. The signal in each full individual channel was adjusted by contrast enhancement (histogram adjusted by saturating

0.35 % of image pixels) to grant proper visualization of each individual channel in the composite merged images. The full display lookup tables (LUT) of each image channel can be accessed on the image repository associated with this work (https://doi.org/10.5061/dryad.vq83bk3z8).

Quantitative assessment of fluorescent intensity signals was done on unprocessed images. From each volumetric scan included in this study, we extracted the following quantitative information for each individual epithelial tubular segment (number of segments for each recording day and group summarized in Supplementary Table 4): (1) Segment identity (proximal tubule (PT) S1 and S2, and distal convoluted/collecting duct tubules (DCT/CD), which we classified based on morphologic differences and raw intensities of FAD and NADH autofluorescence and their ratio (Fig. 2a–c). Out of 875 tubule segments analyzed from partial IRI mice, 8.7% were unclassifiable. (2) Maximal cross-sectional tubular area (μm²). (3) Number of total nuclei/ segment. (4) Number of PI+ nuclei and (5) Number of GFP-expressing cells per segment. Total, PI+, and GFP+ nuclei numbers were assessed in 3D: nuclei counting was performed in a volumetric fashion using the count of 5 planes over a 25 μm volume covering top to bottom of each analyzed segment. Tubule nuclei were excluded from strong blue cytosolic tubule autofluorescence recorded at 750 nm excitation and were thus clearly identifiable as negatively stained objects in individual tubule cells (Supplementary Fig. 3). PI+ nuclei number quantification was performed based on its strong nuclear red emission in necrotic cells in 750 nm track data (Supplementary Fig. 3). GFP quantification was conducted based on 940 nm excitation track data, which is optimal for GFP excitation[55] and allowed for clear separation of GFP signal from tubule epithelium autofluorescence and from granular cast signal, respectively (Supplementary Fig. 3). For GFP count, a specific subgroup was made for cells proliferating within a -25 μm radius from any necrotic cell at day 0. (6) Amount of granular casts (cellular debris) in the tubule lumen was assessed as the area of solid luminal content over the cross-sectional area at the plane of the largest tubule diameter. (7) Morphological classification of tubular remodeling state: flattened simplified epithelium (acute tubule injury), granular casts (acute tubule necrosis), and atrophic tubule was done for each day from surgery in the long-term data set; (8) Classification of tubule segment fate as sham, uninjured, recovered or atrophic was based on morphology assessment at the end of experiment (day 14 or day 21 of long-term data set). Sham included tubular segments from sham controls. Non-necrotic segments were defined as PI-negative at day 0, which remained functionally and morphologically preserved over time. Recovered segments were defined as PI+ on day 0 (>1% of total nuclei) that regained morphologically intact epithelium. Atrophic segments were defined by a collapsed tubular lumen and severely fragmented epithelial appearance (Supplementary Fig. 2). (9) Dynamic PT-S1 function was assessed as albumin reuptake capacity expressed as the mean fluorescent intensity ratio of Alexa594-albumin in the apical cytoplasmic region of PT-S1 segments and in peritubular capillaries ($n = 170$).

In cases of unsuccessful re-identification of individual FOVs during any acquisition time point, serial data from respective time points were missing and could not be included in further analysis. Displayed data state n numbers for analyzed tubule segments and experimental mice, respectively.

For confirmation of PT-S1/PT-S2 segment classification, bolus-tracking time-lapse recordings of i.v. injected 4 kDa dextran conjugated to FITC were analyzed by placing luminal regions of interest (ROIs) in pre-determined PT-S1 and PT-S2 segments. FITC fluorescent intensity in the green light spectrum ($\lambda_{em}$ = 525/50 nm) was extracted from ROIs and plotted over time.

Raw data were collected and assembled in Microsoft Excel v. 2305 for further statistical analysis, as outlined below.

## Image processing

Image processing was done using the recently published "Intravital microscopy processing toolbox for FIJI"[52] using the following features:

Denoising of the Stacks was performed with variance stabilizing transform (VST)[56] BM4D[34] in MATLAB v. 2022a (Mathworks). First, the standard deviation of the noise (sigma) was estimated on 5 separate images with the algorithm from Liu et al.[57] around the middle and the last frames of the total stack and then averaged to a single value. Second, a generalized Anscombe VST[56] was then applied to the data. After applying the VST, the stack was split into 10 frames chunks overlapping by 5 frames at each extremity and then processed in parallel with the BM4D denoising filter. Finally, BM4D-processed chunks were assembled into a single stack, and a closed-form inverse Anscombe VST [36] was applied to obtain the final denoised stack.

Image stacks were registered to enhance the visualization of corresponding focal planes in the presence of severer ischemia-induced tissue remodeling with the use of the FIJI plugin BigWarp and a rigid rotation transform[58]. The day 0 stack of each FOV was set as a target image to use as reference, landmarks were manually placed on follow-up data on corresponding structures. The operation was repeated until the alignment was judged visually successful.

## Animal euthanasia practices

After the last experiment, mice were anesthetized with isoflurane and perfusion-fixated with 4% PFA for tissue collection and subsequent immunohistology.

## Immunohistochemistry

The kidneys were post-fixated in 4% PFA for 2 h at room temperature and then processed for frozen tissue embedding or ex vivo correlative microscopy. For frozen tissue embedding, kidneys were placed in 30% sucrose/PBS until they sank to the bottom, frozen in Tissue-Tek O.C.T. compound (Sakura, USA), and eventually stored at −80 °C. For ex vivo correlative microscopy, kidneys were immediately processed following the staining protocol outlined below.

Correlative microscopy of unstained, HE, and PAS-stained frozen kidney sections: 6 µm thick frozen sections from post-ischemic tissue (n = 3 kidneys) were cut in serial and either stained with hematoxylin and eosin (HE), periodic acid-Schiff (PAS) or left unstained. HE- and PAS-stained sections were imaged on a Leica light microscope, and unstained frozen sections were imaged on an Olympus FVMPE-RS 2-photon microscope using 750 nm excitation and the following detection cubes: Ch1: $\lambda_{et}$ = 570/30 nm, Ch2: $\lambda_{et}$ = 503/35 nm, Ch3: $\lambda_{et}$ = 452,5/22.5 nm. The same tubule segments were localized on HE-, PAS-stained, and unstained sections, respectively, to compare how distinct remodeling events were detected via 2PM displayed in HE- and PAS-stained tissue, respectively. An experienced pathologist reviewed the imaging series and classified different remodeling events as outlined in Supplementary Fig. 2.

Ex vivo correlative microscopy of VCAM-1 immunostained kidney tissue: The cortical renal area, which was attached to the AIW, was separated as a 1–2 mm thick tissue section. The recovered tissue was washed twice in PBS for 15 min, followed by blocking in 1% BSA/2% SEA BLOCK/0,1% Triton X-100/PBS for 1 h. Rabbit anti-VCAM-1 (Abcam AB0134047, 1:200)[12] primary antibody was incubated for 72 h at 4 °C. Subsequently, Donkey anti-Rabbit Alexa 405 (Jackson ImmunoResearch 711-175-152, 1:500) secondary antibody was incubated overnight at 4 °C. After washing with PBS, the tissue section was embedded in PBS between two coverslips separated by a 1–2 mm thick spacer (SunJim Lab) and imaged on an Olympus FVMPE-RS 2-photon microscope. The same tissue regions as imaged in vivo were identified based on tissue landmarks, such as recognizable tubule shapes and fluorescence patterns[32]. Images were acquired at 800 nm excitation wavelength, and emission was collected using the following detection cubes: Ch1: $\lambda_{et}$ = 570/30 nm, Ch2: $\lambda_{et}$ = 503/35 nm, Ch3: $\lambda_{et}$ = 452.5/22.5 nm. Antibody validation: Rabbit anti-VCAM-1 (Abcam AB0134047) specificity for immunohistology in mouse tissue was validated by the manufacturer using knockout validation and mouse spleen tissue as positive control. The suitability of Rabbit anti-VCAM-1 for identifying cells of failed tubule repair was previously established for mouse and human tissue[11,12]. Unspecific staining of Donkey anti-Rabbit Alexa 405 (Jackson ImmunoResearch 711-175-152) was excluded by performing the same staining protocol without primary antibody incubation.

For confirmation of selective GFP expression in proliferating cells, we injected daily 1 µl/g BW of EdU (1 mg/ml in sterile PBS, Invitrogen C10424) i.p. in 3 CycB1-GFP mice for 4 days. Daily serial 2-PM was used to record GFP expression over time. On the 4th day, mice were perfusion-fixated and kidneys post-fixated as described above. Visualization of in vivo injected EdU on fixed tissue was performed using a Click-iT Edu Flow assay Cytometry Assay kit Alexa fluor 647 (Invitrogen, C10424). Samples were washed twice for 15 min in PBS at 300 rpm, 20 °C in an Eppendorf Thermomixer, followed by permeabilization in PBS with 0.5% triton X-100 for 15 min at 300 rpm, 20 °C. Subsequently, samples were incubated in a modified EdU reaction cocktail, where 1% Triton x-100 was added to the PBS for 6 h at 20 °C, 300 rpm. After staining, the sample was washed twice for 5 min in PBS at 20 °C, 300 rpm, and stored in PBS at 4 °C until imaging. To correlate GFP expression as determined in vivo to EdU incorporation as visualized ex vivo, we performed correlative imaging of the same FOVs as imaged in vivo on a Zeiss LSM710 confocal microscope (Zeiss, Germany).

## Statistical analyses

We used MATLAB v. 2022a for statistical analyses (The Mathworks Inc.). For imaging data, individual tubular segments were considered statistical units, and each analysis consisted of a repeated measurement of the same tubule over time. To handle missing data points and the heterogeneous number of samples across different mice, we constructed linear mixed-effects models for each dataset derived from segment analyses. Response variables and fixed effects are specified in "Extended statistical analyses" section in Supplementary Information for each analysis. Results from linear regression analyses are reported as: linear equation (y = slope*x + intercept), $R^2$ goodness-of-fit and effect size estimated as Pearson's correlation coefficient. Predictors and groups are specified in each figure legend. Linear equations are reported on the respective scatterplots to underline statistical comparisons between slopes and intercepts as the effect of a given predictor. Individual animal subjects were used as random effects for all linear mixed-effects models. GFR and Albumin/creatinine ratios were analyzed with a paired two-way ANOVA in Graph Pad Prism (same subject time matched through different days) with Bonferroni correction for multiple comparisons. Quantitative figures and related statistics were made using Gramm toolbox[59]. Data are reported with mean with 95% CI in figures and mean ± SEM in text unless otherwise stated. p-values are reported as following:

*/#: $p < 0.05$; **/##: $p < 0.01$; ***/###: $p < 0.001$. Detailed information on test statistics is reported in the "Extended statistical analyses" section in Supplementary Information, referred to in the text as "Supplementary Extended Statistics", which comprises: For ANOVA and mixed-effects models, F values with degrees of freedom (F (DFn, DFd)) are reported. For differences between groups (in $t$-tests and individual post-hoc groups estimates comparisons from linear mixed models), the model's estimate of the difference between the groups (Δ) with lower/upper 95% confidence intervals (95% CI [LL, UL]) and t values with degrees of freedom (t (df)) are reported.

### Statistics and reproducibility

Intravital imaging experiments of individual experimental mice were performed as independent experiments. Upon reassurance that inclusion criteria were given (successful partial IRI surgery and correct placement of the abdominal imaging window implant), different partial IRI regions (Not-IR, Mid, and IR) were reproducibly identifiable based on distinct necrotic cell damage patterns on day 00. The biological phenomena described in this study were reproducible in all attempted experiments.

Analysis of the albumin-creatinine ratio from urine samples was performed as duplicates. Baseline transcutaneous GFR measurements were performed as duplicates. After IRI/sham surgery, transcutaneous GFR measures were performed longitudinally and were hence obtained as single values per time point and animal.

Representative serial in vivo 2-photon images were selected from data obtained from the following number of independent experiments: partial IRI: day 0 = 8 mice, day 1–2 = 4 mice, day 3 = 8 mice, day 4 = 4 mice, day 7, 21 = 5 mice; day 14 = 4 mice; and sham: day 0, 1, 2, 3, 4, 7, 14 = 3 mice, day 21 = 2 mice. Representative correlative in vivo and ex vivo images for the classification of remodeling processes were selected from data obtained from three independent experiments. Representative correlative in vivo and ex vivo images for the detection of VCAM1-positivity in atrophic tubules were selected from data obtained from three independent experiments.

### Reporting summary

Further information on research design is available in the Nature Portfolio Reporting Summary linked to this article.

## Data availability

The raw data used to generate all the figures shown in this paper are provided as a Source Data file. The two-photon imaging data generated in this study have been deposited in the DataDryad database: https://doi.org/10.5061/dryad.vq83bk3z8. Source data are provided with this paper.

## Code availability

All analysis codes used for statistical analyses and the generation of plots in MATLAB are available in the Zenodo repository: https://zenodo.org/record/7892132.

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

## Acknowledgements

We thank Associate Professor Rikke Nielsen for the fruitful and critical discussion of the data. We further like to thank the Center for Functionally Integrative Neuroscience and Aarhus University Health Bioimaging Core facility for the use of imaging equipment and technical support. I.M.S. has received grants from the Novo Nordisk Foundation (NNF, grant number: NNF19OC0054899) and Aarhus University Research Foundation (AUFF, grant number: AUFF-E-201 9-7-21). L.B. was supported by the European Union's Horizon 2020 Research and Innovation Program under Marie Skłodowska-Curie (grant number: 801133).

## Author contributions

L.B. developed the disease model and designed imaging experiments, acquired in vivo imaging data, analyzed images, curated the dataset, performed statistical analyses, interpreted the data, and wrote the manuscript. A.M.K. acquired in vivo and in vitro imaging data, designed GFR experiments and acquired GFR data, analyzed images, interpreted the data, and wrote the manuscript. D.S. developed the custom-made setup for in vivo imaging, image analysis tools, and the image processing protocol. H.K. processed ex vivo samples, acquired GFR data, performed immunohistochemistry, and laboratory and mouse colony management. L.P.: analyzed images and interpreted the data. S.R.P.K.: Data interpretation. I.M.S. conceptualized the study, supervised the study, funded the work, interpreted the data, and wrote the manuscript.

## Competing interests

The authors declare the following competing interests: Grants were received from the Novo Nordisk Foundation (I.M.S.), the Augustinus Foundation (I.M.S.), the Aarhus University Research Foundation (I.M.S.), and the European Union's Horizon 2020 Research and Innovation Program (L.B.). The remaining authors declare no other competing interests.
