## [Peer Review File · Nature Communications]

Longitudinal tracking of acute kidney injury reveals injury propagation along the nephronREVIEWER COMMENTS

Reviewer #1 (Remarks to the Author):

In the presented manuscript, the Bordoni and Kristensen et al. present the most detailed intravital microscopy assessment of acute kidney injury (AKI) in history. The last authors, Dr. Schiessl, was trained at the laboratory of Janos Peti-Peterdi, a leading figure in IVM, and has now taken this assessment to a novel level. Previously, IVM studies have investigated individual time points, but these authors provide a longitudinal approach by which they can track individual nephrons over weeks. This is possible after surgically introducing an abdominal imaging window (AIW) and subsequently performance of AKI (in this case ischemia-reperfusion injury (IRI), the most widely used model to study AKI mechanisms. The speed of cell death propagation in kidney tubules and the transformation of tubular tissue to fibrotic scarring (referred to as interstitial fibrosis with tubular atrophy (IF/TA) by nephropathologists) are long-standing, yet unanswered questions. By using their technology, they provide very solid evidence for the theory in which cellular necrosis represents the initial pathophysiological event, and that fibrotic changes develop in the very same tubular regions 14 days later. This is a fundamentally important observation because the entire field of renal fibrosis appears to ignore this, assuming that fibrosis and its progression happen without initial damage. This manuscript, based on the undoubtable quality of the longitudinal IVM that allows to look at the very same nephron over weeks, answers this question in vivo.

That said, a number of minor remarks are listed below, entirely meant to help further improve (the presentation of) this work, and maybe the integration into existing literature. Clearly, this manuscript is of interest to the readership of Nature Communications, as this technology can be transferred to most other organs, provides a novel and exciting technology and answers long-standing questions on the origin of tissue fibrosis.

Minor remarks

- The in vivo PI staining (Fig. 1E, Mid) allows to identify the edge of the IRI-damaged region. This is a very interesting area of research, as tubular cell death propagation has been suggested to spontaneously progress in isolated tubules. Did the authors observe live cell death propagation in a tubule that might span over this edge, or is this simply anatomically unlikely?
- Figure 1H is a remarkable figure as it demonstrates how necrosis has affected a part of the kidney tubule, while another part of the same tubule is “still” intact. Given the clarity of this image, I recommend to upload a high-resolution version of it to the supplement of this manuscript. This should be carefully labelled for everything the authors observe here, including PI-positivity, necrosis, an almost unaffected nephron loop (in the center of the necrotic tubule), absence of interstitial fibrosis, but a beginning edema in the interstitial space, S1/S2 segments, and so on.
- In previous reports, the green autofluorescence of the kidney tubules affected the quality of the GFP signals that could be detected over time. How did the authors overcome this? Maybe an additional sentence in the methods section could help.
- The brilliant analysis of GFP-positive nuclei presented in Fig. 3A should start with the day 00 value and progress to day 03, as it was done for Fig. 4A and B!
- In Fig. 3E, this referee recommends to indicate that this is a day 02 picture in the actual figure, not only in the legend.
- The section on the proliferating cells after insult is particularly interesting, but the equation in Fig. 4F is not easy to understand for readers of Nature Communications. What gave rise to this particular method should be explained in some more detail in the results section and in the methods.
- Because of its clinical relevance, I suggest to point out very clearly the first observed cast formation. In many settings on ICUs, muddy brown casts are still interpreted as early indicators of AKI. It would be good for the clinicians to understand that they evolve “late” after the actual AKI/IRI trigger.

Reviewer #2 (Remarks to the Author):

In this translational study, the authors have used repetitive intravital imaging to study the spatiotemporal evolution of cellular events that occur in acute kidney injury (IRI) due to ischemia reperfusion injury (IRI). They report several new findings, including greater injury in the early (S1) part of the proximal tubule (PT), an apparent propagation of injury into later segments, and an association between accumulation of granular casts and tubular atrophy.

A major strength of the study is the impressive technical approach, combining longitudinal intravital imaging with functional markers (e.g. albumin uptake) and transgenic animals expressing a marker of cell cycling. Moreover, the partial IRI model is also innovative. The main limitation is that the findings are largely descriptive in nature, thereby making it difficult to dissect out causation from correlation. This means that whilst the conclusions of the authors are plausible, it's difficult to exclude the possibility of alternative explanations for the findings.

I have the following specific comments:

The major finding of the study is that damage in injured S1 regions subsequently propagates to downstream S2, and the authors conclude that this process is driven by formation of granular casts (although the exact mechanisms underlying this phenomenon are unclear). This conclusion rests mainly on the observation that there is initially much more necrosis and cell cycling in S1, but that cell cycling and tubular atrophy subsequently occurs in S2, especially where granular casts are found. While this is one plausible interpretation of the data, there could be other possibilities. For example, although necrosis is less prominent in S2, cells in this region are presumably also significantly damaged by the initial IRI. The appearance of cell cycling in S2 might therefore be due to the direct effects of IRI, rather than an indirect effect mediated by luminal casts (with the time delay simply reflecting less severe initial injury than in S1). For their analysis, the authors defined regions as "initially undamaged" if they displayed 0% necrotic cell death, but presence of necrosis denotes very severe injury – therefore its absence does not necessarily mean that tubules are undamaged. Moreover, tubular damage in regions of S2 could promote the accumulation of luminal casts, thus explaining the apparent correlation. Ultimately, definitively proving that injury can somehow spread from S1 to S2 would require inducing damage exclusively in the former (e.g. via endosomal uptake of a toxin). Moreover, if granular casts are indeed causing direct injury to S2 cells, it might be expected that an intervention to promote cast removal might be protective.

Following on from this point, the major evidence supporting a relationship between necrotic cells, granular casts and tubular atrophy is a correlative analysis performed on 2-D images. However, this does not account for the 3-D structure of tubules. For example, when the authors observe cell cycling in a non-necrotic S2 region with casts, can they be certain that they are not missing adjacent necrotic cells which are above/below the focal plane?

The authors used a genetically expressed reporter of cell cycling, and from this concluded that necrotic cells are replaced by proliferation of surviving adjacent cells. Moreover, they report that regions of high cell cycling activity become atrophic, rather than recovering. This latter observation is perhaps somewhat surprising, and raises the question as to whether expression of the reporter actually denotes cell proliferation. Do the authors have other evidence of cell proliferation to support their conclusions?

In their figures, the authors provide overlay images of multiple signals, some of which appear to be the same color (e.g. albumin and PI). This makes it difficult to appreciate the individual signal patterns. Therefore, the authors should provide separate images from each channel, especially in the early figures, where signals are being depicted for the first time.

Minor:

When quantifying the number of necrotic (PI positive nuclei), how was the total number of cell nuclei calculated? For in vitro studies, a nuclear marker such as Hoechst is typically used.

While the IRI model is widely used in pre-clinical AKI studies, it remains unclear how relevant it is to human AKI, where tubular cell injury is typically much less prominent (e.g. see PMID: 29933845). This point is underlined by the very high percentage of non-recovering tubules in the ischemic region. The authors could briefly acknowledge this issue in their discussion to place their findings in a wider translational context.

Page 5: "...approximately half of the organ WAS rendered ischemic,"

Page 9: "Of note, closer evaluation demonstrated MARKED differences among segment types"

Page 11: "which over time flowed downstream ALONG the nephron"

Reply to Reviewer 1:

“In the presented manuscript, the Bordoni and Kristensen et al. present the most detailed intravital microscopy assessment of acute kidney injury (AKI) in history. The last authors, Dr. Schiessl, was trained at the laboratory of Janos Peti-Peterdi, a leading figure in IVM, and has now taken this assessment to a novel level. Previously, IVM studies have investigated individual time points, but these authors provide a longitudinal approach by which they can track individual nephrons over weeks. This is possible after surgically introducing an abdominal imaging window (AIW) and subsequently performance of AKI (in this case ischemia-reperfusion injury (IRI), the most widely used model to study AKI mechanisms. The speed of cell death propagation in kidney tubules and the transformation of tubular tissue to fibrotic scarring (referred to as interstitial fibrosis with tubular atrophy (IF/TA) by nephropathologists) are long-standing, yet unanswered questions. By using their technology, they provide very solid evidence for the theory in which cellular necrosis represents the initial pathophysiological event, and that fibrotic changes develop in the very same tubular regions 14 days later. This is a fundamentally important observation because the entire field of renal fibrosis appears to ignore this, assuming that fibrosis and its progression happen without initial damage. This manuscript, based on the undoubtable quality of the longitudinal IVM that allows to look at the very same nephron over weeks, answers this question in vivo.

That said, a number of minor remarks are listed below, entirely meant to help further improve (the presentation of) this work, and maybe the integration into existing literature. Clearly, this manuscript is of interest to the readership of Nature Communications, as this technology can be transferred to most other organs, provides a novel and exciting technology and answers long-standing questions on the origin of tissue fibrosis.”

We thank the reviewer for the positive evaluation of our work!

Minor remarks

- “The in vivo PI staining (Fig. 1E, Mid) allows to identify the edge of the IRI-damaged region. This is a very interesting area of research, as tubular cell death propagation has been suggested to spontaneously progress in isolated tubules. Did the authors observe live cell death propagation in a tubule that might span over this edge, or is this simply anatomically unlikely?”*

We agree that the border region in our partial IRI model is scientifically very interesting. We have studied this region in detail and have found no indication for spontaneous propagation of necrotic cell

death at similar speed as previously shown in isolated tubules. We cannot fully exclude that a similar phenomenon might happen in a slower fashion within the first few hours after reperfusion as our experimental protocol did not include multiple imaging sessions within the first day. Nevertheless, from 1 day after reperfusion, we did observe granular cast accumulation in multiple previously non-necrotic tubule segments in this tissue regions, indicating that individual nephrons may span across the diffuse border of perfused and post-ischemic renal tissue and that cellular debris derived from upstream tubule necrosis accumulate in downstream non-necrotic tubule segments.

• *“Figure 1H is a remarkable figure as it demonstrates how necrosis has affected a part of the kidney tubule, while another part of the same tubule is “still” intact. Given the clarity of this image, I recommend to upload a high-resolution version of it to the supplement of this manuscript. This should be carefully labelled for everything the authors observe here, including PI-positivity, necrosis, an almost unaffected nephron loop (in the center of the necrotic tubule), absence of interstitial fibrosis, but a beginning edema in the interstitial space, S1/S2 segments, and so on.”*

We thank the reviewer for this nice suggestion. Figure 1H has already been provided as a supplemental high-resolution figure on its own with the first submission of this manuscript. As the reviewer suggested, we have now added detailed labeling of the observable events in this image and updated the figure legend accordingly (fig. S9).

• *“In previous reports, the green autofluorescence of the kidney tubules affected the quality of the GFP signals that could be detected over time. How did the authors overcome this? Maybe an additional sentence in the methods section could help.”*

GFP-quantifications were performed on 940 nm excitation track data. 940 nm achieves maximum excitation for GFP with 2-photon, meaning that the signal is the brightest possible. At the same time, green tubule autofluorescence is comparably weak. As also pointed out by reviewer 2, our 2-excitation track (750 nm and 940 nm) images contain a lot of visual information. We therefore decided to provide an additional figure in which we show single track data and explain all detectable signals in detail. In this figure (fig S3), it becomes evident that GFP-signal can be clearly distinguished from tubule autofluorescence. As our analysis protocol aimed for quantification of GFP-positive cell numbers, rather than fluorescence intensity of GFP signal in individual cells, remaining autofluorescence signal in the same detection channel as GFP, did not impose any complications for GFP quantification. As suggested, we also added an additional sentence

about GFP quantification in the methods: *“GFP quantification was conducted on 940 nm excitation track data, which is optimal for GFP excitation (Stoltzfuss et al. Sci Rep, 2015) and allowed for clear separation of GFP signal from tubule epithelium autofluorescence and from granular cast signal, respectively (fig. S3).”*

- *“The brilliant analysis of GFP-positive nuclei presented in Fig. 3A should start with the day 00 value and progress to day 03, as it was done for Fig. 4A and B!”*

Figure 3A includes serial imaging data of day 00, 01, 02 and 03 as it was done for Figure 4A and B.

- *“In Fig. 3E, this referee recommends to indicate that this is a day 02 picture in the actual figure, not only in the legend.”*

In Figure 3E, we have now indicated the imaging time point (day 02).

- *“The section on the proliferating cells after insult is particularly interesting, but the equation in Fig. 4F is not easy to understand for readers of Nature Communications. What gave rise to this particular method should be explained in some more detail in the results section and in the methods.”*

We thank the reviewer for pointing out that this data analysis needs more explanation. The rationale behind the analysis in Fig. 4F was to investigate if there was a linear relationship between the degree of the initial necrotic injury and the degree of the subsequent proliferative response. Furthermore, we aimed to assess if such a relationship differed for PT-S1 and PT-S2 segments, respectively. Thus, we plotted the percent of all PI+ (predictor) and GFP+ nuclei (response) and ran a linear regression analysis to demonstrate that increasing necrotic injury predicted increasing proliferative responses in a way that could be described by a statistically significant linear relationship. Conventionally, results of linear regressions are reported as beta coefficient (the slope of a regression) with associated standard error, R^2 and/or r values, F -values, associated degrees of freedom, and p values for statistical significance. We have reported these values and presented them in equation format. The main finding of fig. 4F is that the intercept of the linear regression equation of PT-S2 is significantly higher than that of PT-S1, which demonstrated a PT-S2 segment population that heavily proliferated in the absence of initial necrotic injury. In contrast, the intercept of the PT-S1 linear regression equation is not statistically different from zero, confirming that proliferation in this segment strictly depends on the presence of initial necrotic injury. Furthermore, we report that there is no

difference between the slopes (beta coefficient) of the of PT-S1 and PT-S2 linear regressions, indicating that once necrotic injury is present, a certain degree of necrosis triggered a comparable proliferative response in both segment types. To outline this better to the reader, we have introduced the following changes:

Result section-

The following paragraph:

„Thus, we performed a linear regression of PI-positive nuclei (% of total nuclei) observed on day 0 and the cumulative number of GFP-positive nuclei (% of total nuclei) observed on days 1 to 3 after partial IRI. For both PT-S1 and PT-S2 segments, a significant correlation was detected (fig. 4F). Of note, there was no significant difference in the slopes of the linear regressions, indicating that a certain degree of necrotic cell death was associated with a comparable proliferative response in PT-S1 and PT-S2 segments (fig. 4F). However, we detected a significantly higher intercept of the PT-S2 linear regression as compared to that of PT-S1.“

Has been changed to:

*„ Thus, we performed a linear regression of PI-positive nuclei (% of total nuclei) observed on day 0 and the cumulative number of GFP-positive nuclei (% of total nuclei) observed on days 1 to 3 after partial IRI. For both PT-S1 and PT-S2 segments, a significant correlation **between the initial number of necrotic cells and the subsequent proliferative response** was detected (fig. 4F). Of note, there was no significant difference in the slopes of the linear regressions, indicating **a comparable necrosis-induced proliferative response in PT-S1 and PT-S2 segments** (fig. 4F). **However, we detected a significantly higher intercept in the linear regression of PT-S2 segments (fig. 4F), which demonstrated a population of PT-S2 segments that proliferated in absence of preceding necrotic injury. In contrast, the intercept of the PT-S1 linear regression equation was not statistically different from zero, confirming that proliferation in this segment strictly depended on the presence of initial necrotic injury.**“*

Methods section-

The following sentence:

„Linear regressions are reported as linear fit with goodness-of-fit and effect size estimated as Pearson’s correlation coefficient“

Has been expanded to:

*„ Results from linear regression analyses are reported as: linear equation ($y = \text{slope} * x + \text{intercept}$), R^2 goodness-of-fit and effect size estimated as Pearson’s correlation coefficient. Predictors and groups are*

specified in each figure legend. Linear equations are reported on the respective scatterplots to underline statistical comparisons between slopes and intercepts as effect of a given predictor. “

- “Because of its clinical relevance, I suggest to point out very clearly the first observed cast formation. In many settings on ICUs, muddy brown casts are still interpreted as early indicators of AKI. It would be good for the clinicians to understand that they evolve “late” after the actual AKI/IRI trigger.”*

To clearly point out the first detection of granular casts (day 1), as suggested in the last comment from this reviewer, we have added a quantification of granular cast abundance from day 0 to 3 in figure 5 (fig. 5C) and further indicated this also in the Results:

„From day 1 after reperfusion, we observed pronounced luminal granular cast formation (fig. 5C), which over time flowed downstream along the nephron (fig. 4B, movie S1) and eventually also appeared in the lumen of previously non-necrotic nephron segments (fig. 4B, 5A).“

Reply to Reviewer 2:

“In this translational study, the authors have used repetitive intravital imaging to study the spatiotemporal evolution of cellular events that occur in acute kidney injury (IRI) due to ischemia reperfusion injury (IRI). They report several new findings, including greater injury in the early (S1) part of the proximal tubule (PT), an apparent propagation of injury into later segments, and an association between accumulation of granular casts and tubular atrophy.

A major strength of the study is the impressive technical approach, combining longitudinal intravital imaging with functional markers (e.g. albumin uptake) and transgenic animals expressing a marker of cell cycling. Moreover, the partial IRI model is also innovative. The main limitation is that the findings are largely descriptive in nature, thereby making it difficult to dissect out causation from correlation. This means that whilst the conclusions of the authors are plausible, it’s difficult to exclude the possibility of alternative explanations for the findings.”

We thank the reviewer for the overall positive evaluation of our work and for raising important and constructive critique. We have addressed all the points raised carefully and believe that they have added great value to our study.

Major points:

“The major finding of the study is that damage in injured S1 regions subsequently propagates to downstream S2, and the authors conclude that this process is driven by formation of granular casts (although the exact mechanisms underlying this phenomenon are unclear). This conclusion rests mainly on the observation that there is initially much more necrosis and cell cycling in S1, but that cell cycling and tubular atrophy subsequently occurs in S2, especially where granular casts are found. While this is one plausible interpretation of the data, there could be other possibilities. For example, although necrosis is less prominent in S2, cells in this region are presumably also significantly damaged by the initial IRI. The appearance of cell cycling in S2 might therefore be due to the direct effects of IRI, rather than an indirect effect mediated by luminal casts (with the time delay simply reflecting less severe initial injury than in S1). For their analysis, the authors defined regions as “initially undamaged” if they displayed 0% necrotic cell death, but presence of necrosis denotes very severe injury – therefore its absence does not necessarily mean that tubules are undamaged. Moreover, tubular damage in regions of S2 could promote the accumulation of luminal casts, thus explaining the apparent correlation. Ultimately, definitively proving that injury can somehow spread from S1 to S2 would require inducing damage exclusively in the former (e.g. via endosomal uptake of a toxin). Moreover, if

granular casts are indeed causing direct injury to S2 cells, it might be expected that an intervention to promote cast removal might be protective.”

We agree with the reviewer that initially non-necrotic tubule segments could have still endured sublethal ischemia-induced injury. For a more correct definition, we have changed the terms “damaged” and “undamaged” to “necrotic” and “non-necrotic”, respectively throughout the manuscript. Furthermore, we performed additional experiments to confirm our hypothesis that granular casts deriving from upstream necrotic injury induce injury propagation with proliferation in previously uninjured tubule segments downstream. As the reviewer suggested, we induced selective injury in S1 proximal tubules and investigated subsequent remodeling processes in truly uninjured S2 segments. To harm S1 exclusively, we used the 2-photon laser as a micromanipulator to specifically target and ablate a pre-defined number of cells in a single S1 proximal tubule segment in the field of view. Propidium iodide staining confirmed selective necrosis in the targeted tubule segments. The results from this experiment confirmed 2 major findings from our AKI dataset: First, 1 day after injury, we detected pronounced proliferation in neighboring cells of necrotic cells in laser targeted S1 segments. Second, we observed subsequent granular cast accumulation in some surrounding S2 proximal tubules, which occurred alongside with proliferation of these tubule segments. Of note, surrounding S2 segments that did not reveal granular cast accumulation did not respond with increased proliferation, even if they had been located immediately adjacent to the laser damaged S1 segment. These observations suggest that proliferation in surrounding S2 segments was not induced by any sublethal damage from the laser injury, as in this case proliferation of surrounding tubules should have been strongest in the immediate proximity of the laser-injury. Instead, we observed proliferative activity only in the presence of granular cast accumulation and often in random locations from the laser-induced injury, indicating that luminal cast-accumulating S2 segments belonged to the same nephron as the laser-targeted S1 segment upstream. The new data set has been integrated into the Results and respective data is demonstrated in a new supplemental figure (fig. S6). We believe that this experiment demonstrated causality between the two correlating phenomena we originally described (granular cast accumulation and subsequent proliferation) and thank the reviewer for this great suggestion. If granted by the journal, we would suggest presenting these results in an additional main figure (in which case we would exceed the maximum figure number allowed by the editorial guidelines). Finally, we fully agree that preventing or removing granular casts would be the optimal intervention and should surely demonstrate protective effects. However, we currently know too little about the molecular nature of granular casts and how they may interact with uninjured tubule epithelium. For a thorough molecular characterization of granular casts and possible interaction pathways with resident tubule cells, we plan using omics techniques that will identify pharmaceutical intervention strategies. However, we feel that these experiments would exceed the scope of our current study, which is centered

around the characterization of injury dynamics in the renal tubule over time which has never been demonstrated before.

“Following on from this point, the major evidence supporting a relationship between necrotic cells, granular casts and tubular atrophy is a correlative analysis performed on 2-D images. However, this does not account for the 3-D structure of tubules. For example, when the authors observe cell cycling in a non-necrotic S2 region with casts, can they be certain that they are not missing adjacent necrotic cells which are above/below the focal plane?”

Quantification of necrotic cells, GFP-expressing cells, and the total number of cells per segment was performed in 3D. Only analysis of the granular casts was performed in 2D (at the widest tubule diameter and normalized to the tubule area at the same optical section of the z-stack). We acquired z-stacks through the entire tubule cross-section with 1 μm step size and analyzed respective cell numbers across the entire volume of the stack. We are thus sure to not have overlooked any necrotic cells if present. We thank the reviewer for pointing out that this important detail of our analysis did not come across in the previously submitted version of the manuscript. Details on PI+ and GFP+ nuclei quantification analysis procedures were only mentioned in the methods but the fact that analysis was actually performed in 3D was not mentioned anywhere else. To make this clearer, we have now added notes in the respective figure legends to clarify that these numbers were assessed in 3D.

“The authors used a genetically expressed reporter of cell cycling, and from this concluded that necrotic cells are replaced by proliferation of surviving adjacent cells. Moreover, they report that regions of high cell cycling activity become atrophic, rather than recovering. This latter observation is perhaps somewhat surprising, and raises the question as to whether the expression of the reporter actually denotes cell proliferation. Do the authors have other evidence of cell proliferation to support their conclusions?”

To investigate proliferating cells *in vivo*, we have used a previously published transgenic reporter model (Klochender et al. Dev Cell. 2012). In this previous work, the reporter was validated in detail and identified GFP-expressing cells as proliferating cells through *in vivo* injections and subsequent staining of BrdU and co-localization studies of GFP with immunostaining against Ki67 and PCNA. To confirm these previous findings, we have now performed *in vivo* EdU injections during laser-induced selective PT-S1 injury. *Ex vivo* EdU visualization and correlative imaging of the same kidney areas as previously imaged *in vivo* demonstrated that GFP-expressing proximal tubule cells incorporated EdU and where thus in fact

proliferating. A new supplemental figure has been provided (fig. S4) and these results are now further outlined in the Results. Indeed, it might be puzzling that increased proliferation was associated with a higher probability of tubule atrophy. However, this observation adds to the novelty of our study. Our longitudinal study demonstrated for the first time that epithelial proliferation occurs at higher rates upon high initial necrosis. Thus, pronounced necrotic injury and subsequent injury propagation may provoke proliferation in a dose-dependent manner that will still render into failed repair and tubule atrophy whenever the initial insult was too severe.

“In their figures, the authors provide overlay images of multiple signals, some of which appear to be the same color (e.g. albumin and PI). This makes it difficult to appreciate the individual signal patterns. Therefore, the authors should provide separate images from each channel, especially in the early figures, where signals are being depicted for the first time.”

We fully agree that our imaging data contains a high amount of biological information. To better convey this information to the reader, we have thus prepared a new supplementary figure (fig. S3), in which we split the two excitation tracks and show individual channel data. Thus, we show how tubule autofluorescence was used to count total number of nuclei in 3D via negative staining, point out how GFP signal was separated from tubule autofluorescence (point raised by reviewer 1), and indicate albumin signal in peritubular capillaries and apical membranes of proximal tubules. To demonstrate PI+ (only detectable on day 00 after partial IRI) and GFP+ nuclei (most abundant on day 03 after partial IRI), we present representative images from day 00 and day 03 time points respectively. It is however not possible to separate alexa594-albumin and PI signal without the use of advanced image processing (e.g. image segmentation) as both alexa594-albumin and propidium iodide are red emitting fluorescent dyes that were excited with 750 nm and were thus detected in the same detection channel. In the figure, we clearly indicated the different subcellular locations of PI and alexa495-albumin signal in renal tubules that allow to distinguish them, nevertheless.

Minor points:

“When quantifying the number of necrotic (PI positive nuclei), how was the total number of cell nuclei calculated? For in vitro studies, a nuclear marker such as Hoechst is typically used.”

As demonstrated in the new supplementary figure S3, the strong blue and cytosolic tubule autofluorescence collected at 750 nm excitation, excludes the nuclei and thereby identifies them as negatively stained objects. When quantifying the total number of nuclei, we took advantage of this. We have further added more information to the methods to explain this approach in more detail:

„Total, PI+, and GFP+ nuclei numbers were assessed in 3D: Nuclei counting was performed in a volumetric fashion using the count of 5 planes over a 25µm volume covering top to bottom of each analyzed segment. Tubule nuclei were excluded from strong blue cytosolic tubule autofluorescence recorded at 750 nm excitation and were thus clearly identifiable as negatively stained objects in individual tubule cells (fig. S3). PI+ nuclei number quantification was performed based on its strong nuclear red emission in necrotic cells in 750 nm track data (fig. S3). GFP quantification was conducted based on 940 nm excitation track data, which is optimal for GFP excitation [53] and allowed for clear separation of GFP signal from tubule epithelium autofluorescence and from granular cast signal, respectively (fig. S3).“

Using a nuclear stain would have been an option, but we restrained from adding even more color to our imaging data. Classic options such as DAPI or Hoechst33342 are further blue emitting and might have interfered with blue tubule autofluorescence making it more difficult to clearly separate and identify the nuclei. Lastly, we worried that repeated in vivo imaging with DNA-staining could introduce light-induced DNA damage.

“While the IRI model is widely used in pre-clinical AKI studies, it remains unclear how relevant it is to human AKI, where tubular cell injury is typically much less prominent (e.g. see PMID: 29933845). This point is underlined by the very high percentage of non-recovering tubules in the ischemic region. The authors could briefly acknowledge this issue in their discussion to place their findings in a wider translational context”

We agree with the reviewer that the relevance of cell necrosis in human AKI is controversially discussed. We had already outlined this fact in the first submission but tried to point this out even more clearly with the revised version of the manuscript:

“Of note, we also observed how moderately necrotic PT-S1 segments presented later without classic signs of ATN (e.g. granular cast formation), but rather displayed a phenotype of ATI, with simplified and thinned epithelium, which lacked brush border lining (fig. 4A D.2-3 and fig. 7A Mid D.3). Since AKI patient biopsies often rather display features of ATI, the role of ATN in human AKI is controversially discussed [22, 42]. Our longitudinal approach allowed following the same renal tubules over time and demonstrated that ATI can be preceded by mild to moderate tubule necrosis. Biopsies from AKI patients are not routinely collected, limited in number, and taken at heterogeneous time points after the insult [22], making it difficult to exclude a role of

necrotic cell death in human AKI. Furthermore, urinalysis from AKI patients demonstrated muddy brown casts with numerous renal epithelial cells, suggestive of ATN [43]. These observations indicate the need for more patient material to re-assess the role of ATN in human AKI.

“Page 5: “...approximately half of the organ WAS rendered ischemic,”

Page 9: “Of note, closer evaluation demonstrated MARKED differences among segment types”

Page 11: “which over time flowed downstream ALONG the nephron””

We thank the reviewer for pointing out the grammatical issues on page 5, 9 and 11. It has all been corrected.

REVIEWERS' COMMENTS

Reviewer #1 (Remarks to the Author):

I went through the paper once again, and I can only congratulate the authors to this significant contribution. I do not recommend any further edits. This work should be made publically available immediately.

Reviewer #2 (Remarks to the Author):

The authors have provided appropriate responses to the issues raised and have added new data to support their conclusions.

The association between activation of cell proliferation and tubular atrophy is still a little puzzling/counter-intuitive. It could therefore be helpful for readers if the authors could speculate briefly in their discussion on possible explanations for this phenomenon.

I have no further comments.

Reply to reviewer 1

“I went through the paper once again, and I can only congratulate the authors to this significant contribution. I do not recommend any further edits. This work should be made publically available immediately.”

We sincerely thank the reviewer for the positive evaluation of our work.

Reply to reviewer 2

“The authors have provided appropriate responses to the issues raised and have added new data to support their conclusions. The association between activation of cell proliferation and tubular atrophy is still a little puzzling/counter-intuitive. It could therefore be helpful for readers if the authors could speculate briefly in their discussion on possible explanations for this phenomenon. I have no further comments.”

We would like to again thank the reviewer for raising good concerns that helped us to improve our manuscript. We are happy that the revised version of the manuscript could meet the reviewer’s expectations. As requested by the reviewer, we have added speculations on possible explanations for the increased cell proliferation observed in atrophying tubules in the discussion. Thus, the following section in the discussion:

“Of note, accelerated epithelial proliferation was not associated with successful recovery (movie S1). On the contrary, we detected a higher number of cycling cells in tubules, which eventually turned atrophic than in those which recovered.

Importantly, granular cast accumulation did not only associate with progressive epithelial proliferation but predicted tubule atrophy development in a dose-dependent fashion. Overall, these observations indicate a previously underestimated role of early proximal tubule necrosis in AKI and for the first time demonstrate evidence for substantial and fate-determining injury propagation along the nephron. Consistent with a concept of necrosis-induced injury propagation into downstream nephron segments, old light and electron microscopy data from post-ischemic rat kidneys illustrated necrosis in proximal convoluted tubules, while proximal straight tubule segments were entirely

blocked by accumulated luminal blebs [35]. Furthermore, membrane blebbing and granular cast formation during AKI, have been associated with tubule obstruction and backflow [34, 35, 40], which may favor tubule atrophy. Lastly, membrane rupture during necrotic cell death forms cellular debris and leads to extracellularization of so-called danger-associated molecular patterns (DAMPs), which may further drive tissue inflammation and injury, a process commonly referred to as necroinflammation [41].“

Has now been modified the following way:

Of note, accelerated epithelial proliferation was not associated with successful recovery (movie S1). On the contrary, we detected a higher number of cycling cells in tubules, which eventually turned atrophic than in those which recovered. **The increased likelihood of tubule atrophy in the presence of enhanced epithelial proliferative activity may be surprising. Nevertheless, our data indicated a linear relationship between the initial degree of tubular necrotic cell death and the number of proliferating tubule cells arising thereafter. Thus, the increased proliferative activity observed in atrophying tubules was also preceded by a higher degree of epithelial necrosis and formation of granular casts. Consistent with this,** granular cast accumulation did not only associate with progressive epithelial proliferation but predicted tubule atrophy development in a dose-dependent fashion. Overall, these observations indicate a previously underestimated role of early proximal tubule necrosis in AKI and demonstrate evidence for substantial and fate-determining injury propagation along the nephron. **Even though these processes still initiated epithelial repair programs, the latter were eventually deemed unsuccessful in the presence of too severe epithelial necrosis.** Consistent with a concept of necrosis-induced injury propagation into downstream nephron segments, old light and electron microscopy data from post-ischemic rat kidneys illustrated necrosis in proximal convoluted tubules, while proximal straight tubule segments were entirely blocked by accumulated luminal blebs [35]. Furthermore, membrane blebbing and granular cast formation during AKI, have been associated with

tubule obstruction and backflow [34, 35, 40], which may favor tubule atrophy. Lastly, membrane rupture during necrotic cell death forms cellular debris and leads to extracellularization of so-called danger-associated molecular patterns (DAMPs), which may further drive tissue inflammation and injury, a process commonly referred to as necroinflammation [41].